# iFISH is a publically available resource enabling versatile DNA FISH to study genome architecture

Eleni Gelali [1], Gabriele Girelli [1], Masahiro Matsumoto [2], Erik Wernersson[1], Joaquin Custodio[1], Ana Mota[1], Maud Schweitzer[1], Katalin Ferenc[1], Xinge Li [1], Reza Mirzazadeh[1], Federico Agostini[1], John P. Schell[3,4,5], Fredrik Lanner[3,4,5], Nicola Crosetto [1] & Magda Bienko [1]

DNA fluorescence in situ hybridization (DNA FISH) is a powerful method to study chromosomal organization in single cells. At present, there is a lack of free resources of DNA FISH probes and probe design tools which can be readily applied. Here, we describe iFISH, an open-source repository currently comprising 380 DNA FISH probes targeting multiple loci on the human autosomes and chromosome X, as well as a genome-wide database of optimally designed oligonucleotides and a freely accessible web interface (http://ifish4u.org) that can be used to design DNA FISH probes. We individually validate 153 probes and take advantage of our probe repository to quantify the extent of intermingling between multiple heterologous chromosome pairs, showing a much higher extent of intermingling in human embryonic stem cells compared to fibroblasts. In conclusion, iFISH is a versatile and expandable resource, which can greatly facilitate the use of DNA FISH in research and diagnostics.

[1] Science for Life Laboratory, Department of Medical Biochemistry and Biophysics,  Karolinska Institutet, SE-17165 Stockholm, Sweden. [2] R&D division, Medical Business Group, Sony Imaging Products & Solutions, Inc., Tokyo 108-0075, Japan. [3] Department of Clinical Science, Intervention and Technology, Karolinska Institutet, SE-14186 Stockholm, Sweden. [4] Ming Wai Lau Centre for Reparative Medicine, Stockholm node, Karolinska Institutet, SE-171 77 Stockholm, Sweden. [5] Division of Obstetrics and Gynecology, Karolinska Universitetssjukhuset, SE-14186 Stockholm, Sweden. These authors contributed equally: Eleni Gelali, Gabriele Girelli. These authors jointly supervised this work: Nicola Crosetto, Magda Bienko. Correspondence and requests for materials should be addressed to N.C. (email: nicola.crosetto@ki.se) or to M.B. (email: magda.bienko@ki.se)

In the past decade, DNA FISH techniques have become increasingly popular among genome biologists, thanks to the fact that they enable direct observation of the three-dimensional (3D) genome architecture in a manner that is complementary to chromosome conformation capture methods, such as Hi-C[1]. Indeed, DNA FISH is now widely accepted as the primary methodology for the validation of Hi-C results. In recent years, several studies have applied DNA FISH combined with super-resolution microscopy to visualize chromosome territories[2–4], folding of chromatin in different epigenetic states[5], topologically associating domains[6–9], and the 3D topology of selected loci in single cells[10].

Historically, DNA FISH probes have been produced by cloning chromosomal fragments of interest into bacterial artificial chromosomes (BACs), followed by their labeling with fluorescent dyes[11]. This approach is time-consuming and logistically challenging. Another limitation is the relatively large size that BAC probes span, which precludes visualizing DNA loci only few kilobases in size. To overcome these limitations, DNA FISH probes composed of chemically synthesized oligonucleotides (oligos) were introduced, and shown to enable high-resolution detection of genomic loci spanning only few kilobases (kb)[12,13]. Soon after, we developed high-definition DNA FISH (HD-FISH), a versatile method for producing DNA FISH probes consisting of double-stranded DNA amplicons generated by PCR[2]. We showed that HD-FISH probes can be used to localize individual genomic loci at sub-diffraction limit resolution, as well as to visualize chromosomal territories as discrete clouds of fluorescence spots—an approach which we named 'chromosome spotting'[2], that allows to quantitatively assess various structural properties of chromosomes. At the same time, another group reported Oligopaint FISH[3], a versatile method to produce DNA FISH oligo probes, named Oligopaints, which can be used to visualize entire chromosomes or sub-chromosomal regions. Several protocols for the enzymatic production of thousands of oligo species starting from pools of chemically-synthesized oligos (oligo-pools) are now available[3,14–17]. Considering that oligo probes bind their target more efficiently compared to double-stranded BAC or amplicon-based DNA FISH probes, oligo probes, such as Oligopaints, are now regarded as the probe type of choice for DNA FISH applications to assess 3D genome architecture at high resolution.

Despite the fact that recent technological improvements have revitalized the application of DNA FISH techniques in the study of genome organization, the community still lacks a comprehensive and standardized resource of probes targeting a large number of defined genomic loci. At the same time, there are no user-friendly computational tools that can be readily used to design DNA FISH probes. Software, such as OligoArray[18] and PROBER[19], have been used to design DNA and RNA FISH oligo probes, including Oligopaints, but their use requires local installation and dedicated bioinformatic expertise. More recently, a freely accessible in silico pipeline, named OligoMiner, was introduced for rapid, genome-scale design of Oligopaints[20]. While this represents a very valuable tool, the OligoMiner databases are the only available ready-to-use databases of oligos for DNA FISH and, as such, leave space for further improvements. One immediate improvement would be increasing the percentage of the genome covered by oligos, given that OligoMiner databases cover the human genome relatively sparsely. Furthermore, while OligoMiner supports the design of oligo databases, it does not provide tools for selecting specific sets of oligos within a given genomic region of interest. In particular, OligoMiner cannot be used for more elaborate probe design strategies that take into account the density of oligos within a given genomic region of interest, and/or the homogeneity of the distribution of the distances between consecutive oligos in the region. This is particularly relevant when designing probes targeting short (10–20 kb) genomic regions, as well as for designing chromosome-spotting probes composed of multiple probes evenly spaced on the same chromosome. In addition, at present, there is no publicly available repository of individually tested and validated probes against multiple loci on different chromosomes. Such repository would enormously facilitate the use of DNA FISH among genome biologists and, importantly, help establish standardized probe collections, which in turn would improve inter-laboratory reproducibility.

Toward this goal, we establish a database of oligos supporting DNA FISH probes design, and targeting a substantially larger fraction of the human genome than the available databases. Most importantly, we create a large resource of validated DNA FISH oligo probes and probe design tools—which we name iFISH—that can be freely used by individual researchers, as well as diagnostic laboratories, to visualize multiple regions of the genome at high resolution, as well as to design probes in regions currently not covered by our repository. We provide a user-friendly web interface (http://ifish4u.org) that can query available curated databases of oligo sequences, and design probes of variable size and number along the genome of interest. Furthermore, we describe a continuously expanding repository of probes—now containing 380 probes in multiple colors targeting 380 loci on all the human autosomes and chromosome (chr) X—which we make publically available. We expand the repertoire of colors that are typically used in DNA FISH, and show that iFISH probes can simultaneously visualize six distinct genomic loci on the same chromosome, labeled with different fluorescent dyes. Lastly, we show that iFISH probes can be used as chromosome-spotting probes to visualize multiple pairs of chromosomes in the same cells, and propose an analytical approach to quantify the extent of intermingling between heterologous chromosomes.

## Results

**Implementation of iFISH.** In order to enable versatile design of DNA FISH probes, we initially developed an algorithm that, given a genome-wide database of oligos and a genomic region of interest, identifies the most suitable window(s) within that region of interest and designs FISH probe(s) by selecting from the database the oligos that map within that window(s), based on various parameters that we have identified as important. Briefly, for a single genomic region, the oligos are selected based on: (i) probe size, i.e., the difference between the genomic coordinates of the 5′-most and 3′-most oligo in the probe; (ii) probe homogeneity, i.e., how homogeneously the oligos are distributed within each probe; and (iii) probe centrality, i.e., how close the probe midpoint is to the midpoint of the specified genomic region (Fig. 1a and Methods). Probe size can range from few kb to megabases (Mb), depending on the application. Probes of 10–20 kb size are ideal to achieve high-precision localization of the resulting FISH signals, whereas larger probes are more suited for applications in which very high detection sensitivity is required. Probe homogeneity is particularly relevant, as uneven distribution of oligos within a probe may lead to the resulting FISH signal splitting into two or more dots, thus reducing the 3D localization accuracy and adding uncertainty to what is being classified as true signal. In addition, the algorithm can be used to design multiple probes evenly spread along the same chromosome (or a smaller region of interest), which in turn can serve to generate chromosome-spotting probes (Fig. 1a and Methods). The algorithm can be run either locally or remotely through a user-friendly interface, which we have named iFISH4U, and made freely available at http://ifish4u.org.

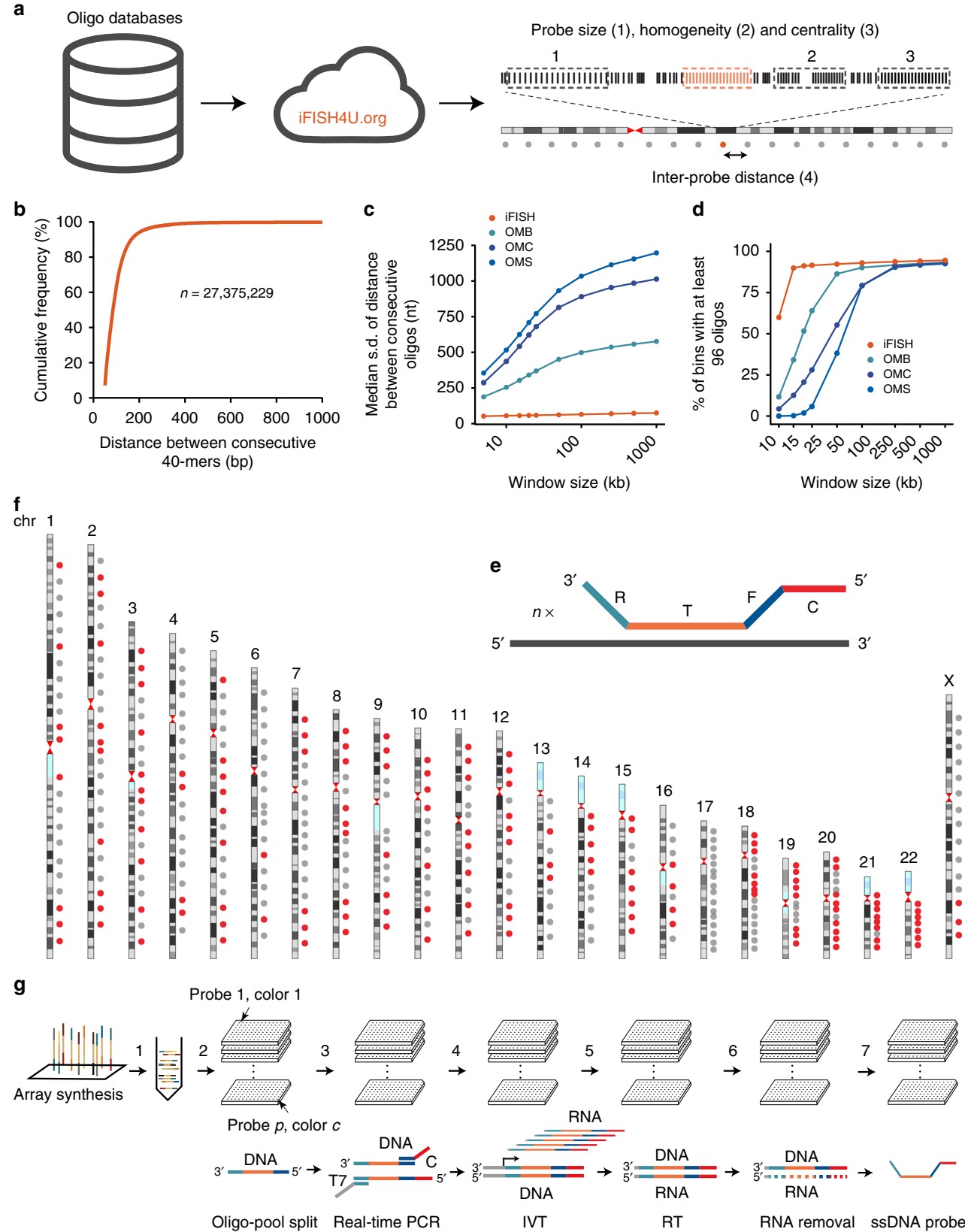

We then sought to establish an oligo database that could be used to design DNA FISH probes for visualizing as many loci as possible, at high resolution, all along the human genome. Such probes would provide a powerful resource for studying 3D genome architecture in single human cells. To this end, we created a high-coverage database consisting of unique, non-overlapping 40-nucleotides (nt)-long oligos (40-mers), extracted from the human reference genome (Methods). We selected 40-mers based on a variety of thermodynamic parameters that are thought to influence the efficiency of in situ DNA hybridization,

**Fig. 1** iFISH implementation. **a** Scheme of iFISH4U. Pre-designed genome-wide databases of oligos (left) are used as input by the iFISH4U web interface (center) to select oligos within one or more user-specified genomic regions, based on the indicated features. Features 1–3 are used while designing single probes, whereas all the four features are used to design multiple probes on the same chromosome. The black dashed boxes indicate examples of probes within the same region of interest, with the same number of oligos (vertical bars), but suboptimal size (1), homogeneity (2), or centrality (3), whereas the orange box represents the probe of choice having optimal size, homogeneity and centrality. **b** Cumulative distribution of the distances between consecutive oligos in the human 40-mers database. **c** Median standard deviation (s.d.) of the distance between consecutive oligos, inside non-overlapping genomic windows of the indicated size, in the 40-mers database and OligoMiner (OM) hg19 databases. OMB, OM 'Balance'. OMC, OM 'Coverage'. OMS, OM 'Stringent'. **d** Percentage of non-overlapping genomic windows of the indicated size, containing at least 96 oligos, in the 40-mers database and OM hg19 databases. **e** Scheme of oligos in iFISH probes. Each probe consists of $n$ oligos differing in the T sequence. **f** Location of the 330 iFISH probes targeting all the human autosomes and chrX. Red dots, individually tested probes (see Fig 2a, b). **g** Scheme of the pipeline used to produce iFISH probes. (1) Up to 12,000 oligos, corresponding to a maximum of 125 probes each containing 96 oligos, are synthesized on an array and then pooled together. (2) The oligo-pool is dispensed into $n$ 96-well plates, depending on the total number of probes ($p$) and colors per probe ($c$). (3) In each well, the oligos corresponding to the same probe are selectively amplified using a probe-specific PCR primer pair that incorporates the T7 promoter sequence (T7) and color adapter sequence (C), and (4) successfully amplified probes are purified and linearly amplified by in vitro transcription (IVT). (5) Purified IVT products are reverse transcribed (RT), (6) RNA is hydrolyzed, and finally (7) single-stranded DNA (ssDNA) is purified to obtain ready-to-use probes

by selecting parameter values similar to those previously used to design Oligopaints (Supplementary Data 1 and Methods). The database can be downloaded at http://ifish4u.org/download, and contains 27,375,229 non-overlapping 40-mers densely covering all the human chromosomes, both in coding and non-coding regions (Fig. 1b, Supplementary Fig. 1a, b). A side-by-side comparison revealed that our database contains 78% more oligos than the OligoMiner (OM) 'Balance' database—the most populated OM database targeting the hg19 human reference genome[20]—and that our database has higher homogeneity of the distances between consecutive oligos, compared to any of the OM hg19 databases (Fig. 1c and Supplementary Fig. 1c–e). Notably, the fraction of 15 kb genomic regions containing at least 96 oligos is ~2.6 times higher in our database compared to the OM 'Balance' hg19 database, corresponding to ~90% and ~34% of the genome, respectively for our and the OM 'Balance' database (Fig. 1d and Supplementary Fig. 1f). These results suggest that our database might be more suited than the OM databases for designing DNA FISH probes against the human genome, especially when targeting small regions.

**Assessment of iFISH probes**. We then assessed how probes designed using our 40-mers database perform in comparison to probes targeting the same regions, but designed with OM. We designed three probes targeting three ~8-kb-sized loci on different chromosomes, either by selecting the oligo sequences from our 40-mers database ('iFISH' probes), or from the OM 'Balanced' hg19 database ('OMB' probes). Each oligo in the probes has a configuration similar to the one previously adopted for oligos in Oligopaints and MERFISH probes[3,21] (Fig. 1e). For each locus, we designed two different OMB probes: (i) one probe consisting of an equal number of oligos (96) as the corresponding iFISH probe ('OMB96'), but with larger size due to the lower coverage of the OMB database, and (ii) one probe of the same size as the corresponding iFISH probe, but with less oligos ('OMB-short') (Supplementary Fig. 2a, Supplementary Data 2 and Methods). Depending on the experimental conditions used, the distributions of FISH dot counts per nucleus were either similar for iFISH and OMB96 probes or closer to the expected distribution in the case of iFISH probes, while the average signal intensity per nucleus was higher in the case of iFISH probes, independently of the conditions used (Supplementary Fig. 2b–g). In contrast, a large proportion of cells hybridized with OMBshort probes showed no signal (35–63% depending on the fluorescence channel), and detectable signals were on average 77% less intense than iFISH signals, many being likely unspecific (Supplementary Fig. 2h, i). The OMB probes performed worse than the iFISH probes despite having a more narrow, and presumably better

distribution of the GC content of their oligos, indicating that it is more beneficial to increase the density of the oligos per kilobase, even if this results in probes with a larger GC content range (Supplementary Fig. 2j).

To further assess the performance of iFISH probes targeting small genomic loci, we created a probe targeting ~14 kb encompassing the *MYC* gene locus (Supplementary Data 3) and performed simultaneous DNA FISH and single-molecule RNA FISH (smFISH[22]) to visualize the *MYC* locus together with its transcripts, in the same cells (Supplementary Fig. 3a, Supplementary Data 3, and Methods). The distribution of DNA FISH dot counts per nucleus showed a major peak at two dots per nucleus, and 85.7% of the cells had 2–4 dots per nucleus, as expected, which is indicative of high specificity (Supplementary Fig. 3b). Furthermore, the transcription site visualized by smFISH (*i.e.*, the site of active *MYC* expression) had a DNA FISH dot, corresponding to the *MYC* locus, in very close proximity in 82.3% of the cells analyzed, which indicates high sensitivity (Supplementary Fig. 3c). Importantly, the efficiency of smFISH was similar when performing the procedure alone or simultaneously with DNA FISH (Supplementary Fig. 3d). We also tested a larger probe (~1 Mb in size) encompassing the human *MYC* locus, which in diploid cells yielded 2–4 large dots per cell in all the cells analyzed, demonstrating that iFISH probes achieve maximum detection specificity and sensitivity when targeting regions in the megabase range (Supplementary Fig. 3e, f, and Supplementary Data 4). Altogether, these results indicate that our oligo database and iFISH4U interface are suitable for designing functional DNA FISH probes, particularly when small genomic regions need to be targeted.

**Creation of the iFISH probe repository**. We then applied the iFISH4U interface, in combination with the newly created oligo database, to design a total of 330 probes targeting 330 loci evenly spaced every ~10 Mb on chr1–16 and chrX, and every ~5 Mb on chr17–22 (Fig. 1f, Supplementary Fig. 4a, Supplementary Data 5, 6, and 7, and Methods). Each probe comprises 96 oligos consisting of four parts (from 5′ to 3′): (i) a 20 nt adapter, C, for probe visualization; (ii) a 20 nt adapter, F, for PCR amplification during probe synthesis; (iii) a 40 nt T sequence complementary to the target; and (iv) a 20 nt adapter, R, for PCR amplification during probe synthesis (Fig. 1e). To design the F and R adapter sequences, we took advantage of a database of non-cross-hybridizing 20-mers, which we derived from a previously published database of 240,000 25-mers orthogonal to the human genome[23] (Supplementary Data 8 and Methods). The median size of the 330 probes is 7.9 kb (range: 7–21.6 kb), with only two probes larger than 10 kb (Supplementary Fig. 4b). Most of the

oligos in the probes (95.93%) map to intronic or intergenic regions, and are therefore highly suitable for the combination of DNA FISH and smFISH (Supplementary Fig. 4c). The genomic coordinates and sequences of the oligos in each probe can be viewed and downloaded at http://ifish4u.org/browse.

We then set out to produce all the 330 probes individually. To this end, we developed a pipeline for large-scale enzymatic production of hundreds of probes in parallel, using 96- or 384-well plates (Fig. 1g). This pipeline builds on the workflow previously described to synthesize Oligopaints and MERFISH probes[3,17,21], introducing several modifications that make the workflow more cost-efficient and compatible with high-throughput liquid handling devices (Methods and Supplementary Notes 1 and 2). A step-by-step protocol can be found in Protocol Exchange[24]. We produced all of the 330 probes in three or more colors, by using different C adapters and unique combinations of F and R adapters. In this way, each probe can be used either as a single probe or combined to other probes in our collection, thus providing extreme flexibility. On average, 78% of all the probes (77% of probes with the C1 and C3 adapters, and 85% of probes with the C2 adapter) were amplified in the first attempt using standard PCR conditions, while the remaining probes were successfully amplified following one or two extra optimization rounds. Independently of the C adapter used, all the probes showed similar amplification kinetics as revealed by real-time PCR (Supplementary Fig. 4d). A melting curve analysis showed a major peak common to all the probes, allowing us to exclude that multiple probes of different length and/or composition were generated in the same well (Supplementary Fig. 4e). To assess the fraction of oligo species within each probe, which are successfully amplified during PCR and in vitro transcription (IVT), we quantified the relative abundance of each oligo in four different, randomly selected probes (Supplementary Data 9 and Methods). Only three oligos in probe 8.13 on chr8, and one oligo in probe 5.16 on chr5, were not amplified in three replicate PCR reactions, which corresponds to a drop-out rate of ~1% (Supplementary Fig. 4f). Overall, these results demonstrate that iFISH enables high-throughput, parallelized production of oligo-based DNA FISH probes of very high complexity.

**Validation of the iFISH probe repository**. We then assessed the performance of the probes in our repository. To this end, we first carried out a large-scale validation effort, in which we individually tested 153 out of 330 probes (46%) in different colors, and analyzed a total of 162,305 FISH dots from 47,747 HAP1 haploid chronic myeloid leukemia cells (Supplementary Fig. 5a). The distributions of dot counts per cell were highly homogeneous across all the probes tested, displaying a major peak at 1–2 dots per cell, as expected (Fig. 2a). The distributions of the mean signal-to-noise ratio (SNR) of the FISH dots in each cell were also similar across all the probes imaged in the same channel (Fig. 2b and Methods). Importantly, the genomic loci targeted by the 330 probes have a DNA accessibility profile which is similar to the rest of the genome (Supplementary Fig. 5b). Accordingly, the dot counts and SNR distributions of the 153 individually tested probes were independent of the DNA accessibility and expression status of their genomic targets (Fig. 2c, d, and Methods). Altogether, these results indicate that the probes in our repository are of homogeneous high quality, independently of the genomic region targeted and combination of adapters used to produce them.

To further increase the versatility and applicability of the probes in our repository, we expanded the repertoire of fluorescent dyes typically used in DNA FISH. Using a custom-built microscopy setup that can reliably discriminate up to seven

fluorescent channels (Supplementary Table 1–3), we imaged six consecutive probes on chr18 together with DNA in HAP1 cells, visualizing each probe in a different color (Fig. 2e–f).

Most of the cells (74% ± 6.5%, mean ± s.d.) had 1–4 dots per nucleus, as expected (Fig. 2g). These results confirm that, independently of the color used, the probes in our repository are able to detect targets of <15 kb with similarly high efficiency. Importantly, the possibility to use six different colors in the same experiment greatly improves the multiplexing capacity of DNA FISH.

**Visualization of chromosome territories**. Having validated our large probe repository, we used it to generate probes for visualizing chromosome territories based on our previously described chromosome-spotting approach[2]. We first visualized chr1, chr6, and chr17, separately, in IMR90 diploid human fibroblasts. This approach revealed distinct clouds of fluorescent spots in each cell, which represent landmarks of the corresponding chromosomal territories (Fig. 3a). On average, 84.7% of cells had ± 20% of the number of FISH dots expected based on the number of probes for the same chromosome (88% for chr1, 86% for chr6, and 80% for chr17), which indicates high sensitivity and specificity of iFISH probes (Fig. 3b). Furthermore, in cells with two clearly distinguishable dot clusters (i.e., territories corresponding to the same chromosome), the number of dots per cluster peaked at the expected value (Supplementary Fig. 6a). We also produced each of the sixteen probes on chr17 separately, and mixed them in equal amounts only before hybridization, obtaining very similar distributions of dot counts per cell (Supplementary Fig. 6b and Methods). Thus, pooling multiple probes targeting the same chromosome before IVT does not diminish the quality of the resulting chromosome-spotting probe, while it significantly speeds up the production of multiple spotting probes.

We then designed 46 extra probes on chr17, and created a chr17-spotting probe consisting in total of 63 probes, in four alternating colors, targeting 63 loci separated on average by 1.25 Mb (Fig. 3c, d, and Supplementary Data 10 and 11). The estimated volume of chr17 territory increased with the number of FISH probes used to visualize the territory, approaching saturation when the probes in all the colors were used to estimate the volume (Fig. 3e and Methods). These results suggest that chromosome-spotting probes consisting of probes separated by ~1 Mb might provide the most accurate estimate of the volumes of the corresponding chromosome territories.

In addition to interphase cells, we assessed the performance of iFISH probes in mitotic nuclei where DNA FISH is notoriously more challenging. To this end, we took advantage of our probe repository to prepare spotting probes in different colors, targeting six different chromosomes, spanning all the range of human chromosome sizes (chr1, chr5, chr10, chr15, chr20, and chrX). To preserve the native nuclear organization of mitotic cells, we applied a 3D DNA FISH protocol which preserves nuclear architecture[25,26] to an asynchronous cell population, and identified mitotic cells based on DNA staining. To increase the number of observable mitotic cells, we imaged all the six chromosomes simultaneously in human embryonic stem cells (hESCs), which cycle very rapidly, and thus have a higher fraction of mitotic cells compared to other cell lines. All the six chromosomes were detected in mitotic nuclei as distinct clouds of dots in different colors, demonstrating the ability of iFISH probes to recognize their target even in highly compact mitotic chromosomes (Fig. 3f). Overall, these results demonstrate that iFISH chromosome-spotting probes are a powerful tool for visualizing chromosomal territories in both interphase and mitotic nuclei.

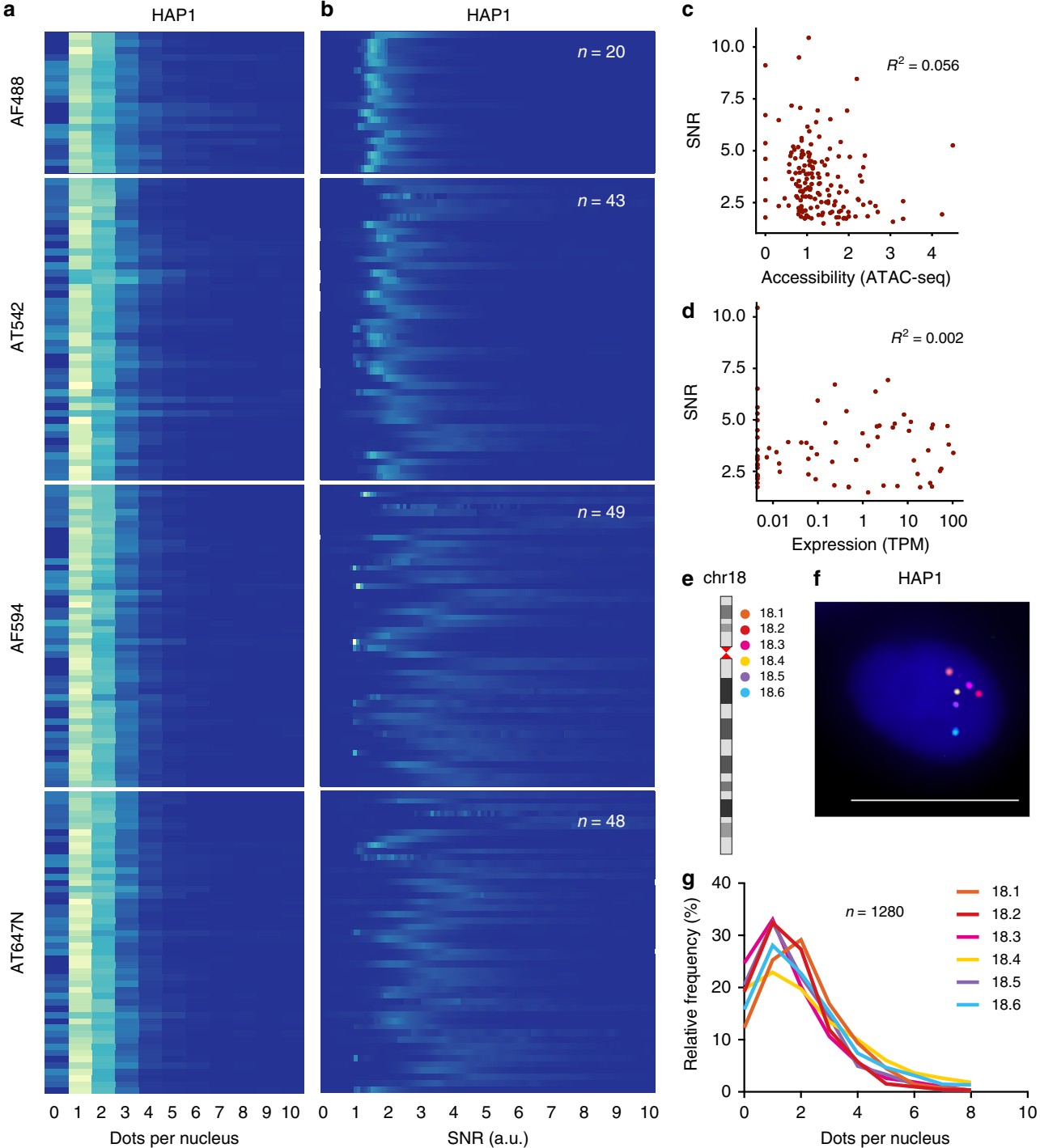

**Fig. 2** Validation of iFISH probes. **a** Distributions of dot counts per nucleus for 153 out of the 330 iFISH probes shown in Fig. 1f, visualized in different colors in G1-phase HAP1 cells. AF488, Alexa Fluor 488 dye. AT542, ATTO 542 dye. AF594, Alexa Fluor 594 dye. AT647N, ATTO 647N dye. n, number of probes visualized in each channel. **b** Distributions of mean signal-to-noise ratios (SNR) per cell for the same probes shown in **a**. **c** Comparison between the mean SNR of each of the 153 probes analyzed in **a** and the DNA accessibility of the corresponding genomic target previously assessed by ATAC-seq[45]. R, Pearson's correlation coefficient. **d** Same as in **c**, but comparing SNR with gene expression levels in the corresponding probe target. **e** Location of six consecutive iFISH probes on chr18, taken from the 330 probes shown in Fig. 1f, that were simultaneously visualized in six different colors. **f** Representative image of the six loci depicted in **e**, visualized in HAP1 cells. Blue, DNA. Scale bar, 10 μm. The maximum intensity z-projection of each channel is shown. **g** Frequency distribution of the number of dots per nucleus, for each of the six iFISH probes in the images of which **f** is a representative example. Probes are numbered according to their location shown in **e**

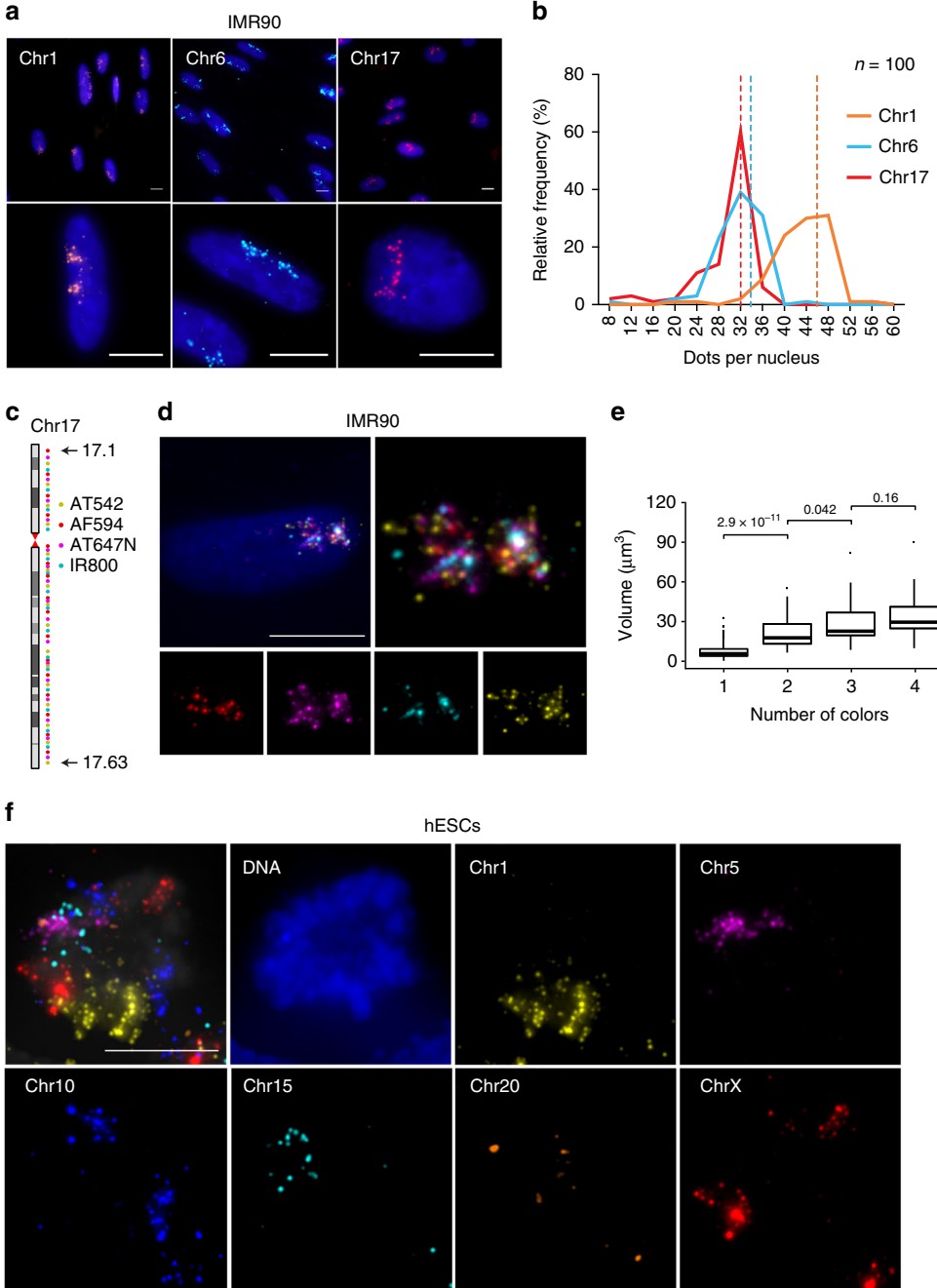

**Fig. 3** Visualization of chromosomal territories by iFISH chromosome-spotting probes. **a** Representative images of chr1, 6, and 17 territories in IMR90 cells. Blue, DNA. Scale bar: 10 μm. **b** Frequency distributions of dot counts per nucleus in the images of which **a** is a representative example. Dashed lines, expected dot counts per nucleus. *n*, number of G1-phase cells analyzed. **c** Scheme of 'dense' chr17-spotting probe consisting of 63 probes spaced every ~1.25 Mb, in four alternating colors. AT542, ATTO 542 dye. AF594, Alexa Fluor 594 dye. AT647N, ATTO 647N dye. IR800, IRDye 800 dye. **d** Examples of chr17 territories visualized using the probes depicted in **c** in HAP1 cells. **e** Chr17 territory volume estimated based on FISH dots in one, two, three, or four colors, using the probes shown in **c**. *P* values (Wilcoxon test, two-tailed) are shown for each of the indicated pair-wise comparisons. In all the box plots, the central line represents the median, the bottom and upper bounds of the box represent the 25th and 75th percentile respectively, and the whiskers extend from −1.5 × IQR to + 1.5 × IQR from the closest quartile, where IQR is the inter-quartile range. **f** Simultaneous visualization of six different chromosomes in mitotic hESCs using chromosome-spotting probes. Scale bar: 10 μm. All the microscopy images in this figure are the maximum intensity z-projection of each channel

**Quantification of chromosome intermingling**. Lastly, we took advantage of our unique way of visualizing chromosome territories using chromosome-spotting probes, aiming to develop a quantitative approach for assessing whether heterologous chromosomes intermingle, which remains a debated issue in the field[27–31]. We first visualized multiple chromosomes in various combinations in IMR90 cells. Typically, in G1 cells, each chromosome appeared as two distinct clouds of dots representing individual chromosome territories, and heterologous territories were often admixed (Fig. 4a). In order to quantify the extent of

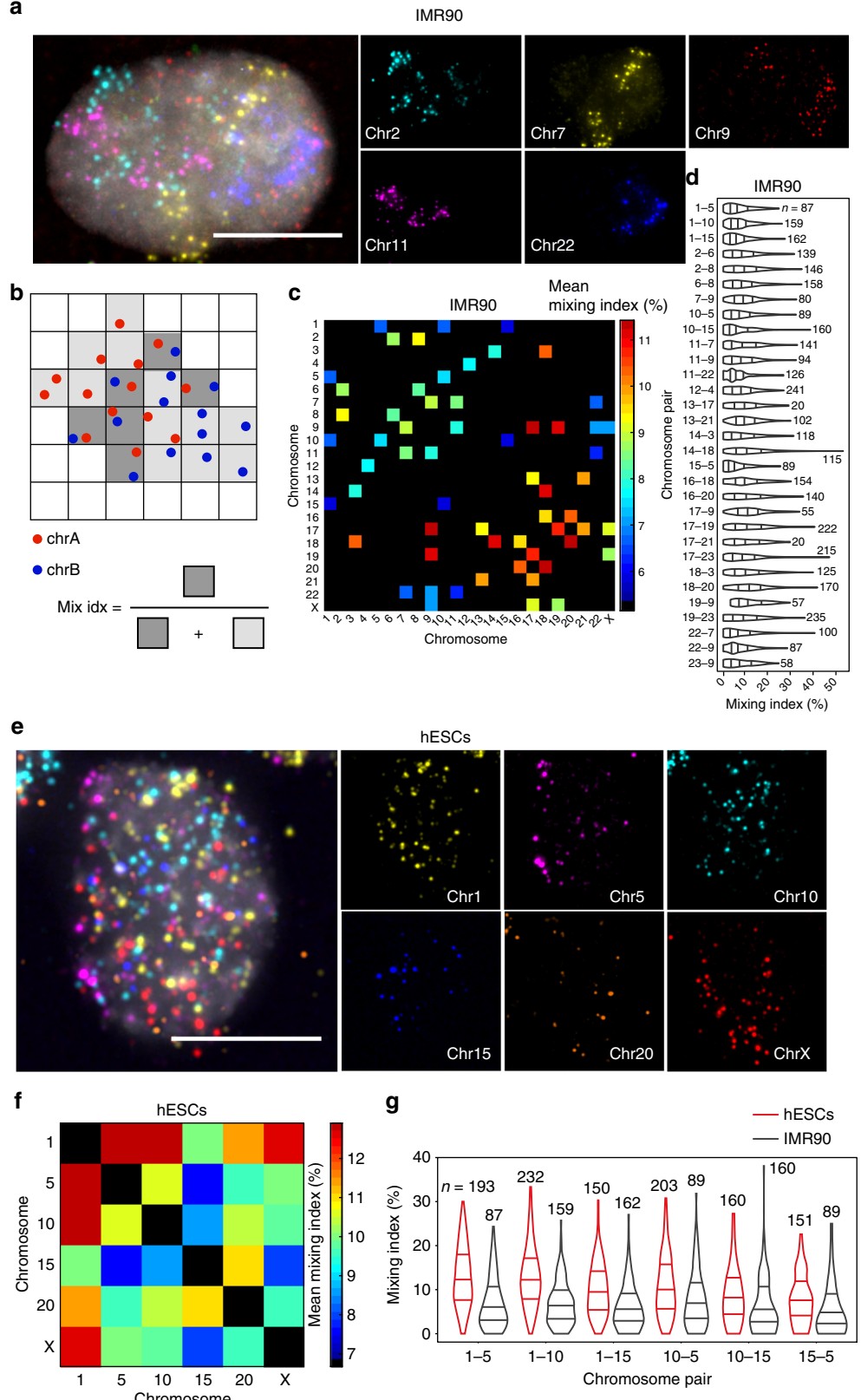

intermingling between heterologous chromosomes, we devised an analytical approach that relies on the dotted nature of the DNA FISH signal produced by chromosome-spotting probes. For a given chromosome pair, we computed a 'mixing index' by dividing the 3D nuclear volume into a regular array of cubes, and then calculating the fraction of cubes containing dots in two different colors (Fig. 4b and Methods). We analysed 31 chromosome pairs, including pairs of similarly sized, long ('L-L') and

**Fig. 4** Quantification of chromosome intermingling using iFISH chromosome-spotting probes. **a** Representative image of the territories of five different chromosomes visualized in IMR90 cells. Gray, DNA. Scale bar: 10 μm. **b** Scheme of how the mixing index is calculated for a given pair of chromosomes, A and B. For simplicity, dots are represented in 2D. The mixing index is the ratio between the number of squares containing dots in both colors and the total number of squares with dots in one or two colors. **c** Average mixing index for 31 chromosome pairs visualized in IMR90 cells. **d** Distributions of mixing index values per cell, in each of the 31 chromosome pairs visualized in **c**. **e** Example of six chromosomes exhibiting complete lack of territoriality and intermingling at a large extent, in the nucleus of one hESC. Gray, DNA. Scale bar: 10 μm. **f** Average mixing index, in hESCs, for six of the chromosome pairs shown in **c**. **g** Comparison between the distributions of mixing index values per cell, for the six chromosome pairs visualized in both hESCs and IMR90 cells. All the microscopy images in this figure are the maximum intensity z-projection of each channel. In all the violin plots, the bottom or leftmost line represents the 25th percentile, the top or rightmost line represents the 75th percentile, and the midline represents the median. *n*, number of cells in which the corresponding chromosome pair was visualized

short ('S-S') chromosomes, as well as pairs of differently sized chromosomes ('L-S'). The average mixing index ranged between 5.2 and 11.4%, and for all the pairs we detected cells in which the mixing index exceeded 20% (Fig. 4c, d). Notably, the mixing index was higher for S-S pairs compared to L-L pairs (Fig. 4c), which is in line with prior reports based on Hi-C measurements[1,32]. Similar results were obtained using a different analytical approach, by computing a 'neighbor index' defined as the fraction of dots in one color that have the nearest neighbor dot in another color (Supplementary Fig. 7a and Methods).

We additionally visualized 15 chromosome pairs in hESCs, including examples of each of the three types of chromosome pairs (L-L, S-S, and L-S). Six out of these 15 pairs were also imaged in IMR90 cells. In hESCs, chromosome territories appeared looser and more admixed compared to IMR90 cells, and, remarkably, we found multiple instances of hESCs with complete lack of chromosomal territoriality (Fig. 4e). In line with these observations, the extent of intermingling between heterologous chromosomes was significantly higher in hESCs, independently of the approach used to quantify it (Fig. 4f, g, and Supplementary Fig. 7b, c). Furthermore, automated K-means clustering of FISH dots performed significantly better in the case of IMR90 cells, indicating that chromosomal territories are on average better defined in these cells (Supplementary Fig. 7d). These results demonstrate that iFISH chromosome-spotting probes can be used to quantify chromosome intermingling in a way that would not be possible with conventional chromosome-painting probes.

## Discussion
For a long time, DNA FISH has represented a rather specialized technique rarely used outside of clinical genetics laboratories, although most of the pioneering work in the field of genome organization was made possible by DNA FISH[33]. With the advent of Hi-C[1], the need to validate and visualize the structural aspects of 3D genome architecture revealed by these methods, directly in single cells, has revived interest in DNA FISH amidst genome biologists. However, despite its undisputable relevance for the field of genome organization, the adoption of DNA FISH as a routine technique for investigating genome architecture has been lagging behind Hi-C and other high-throughput chromosome conformation capture methods[34]. A major reason for this, in our opinion, is the fact that there are no freely accessible repositories of well-curated and ready-to-use DNA FISH probes against a large number of genomic targets. As we have demonstrated in this work, having access to such probe repository dramatically increases the flexibility in the type and number of genomic regions that can be studied by DNA FISH, and consequently the number of applications and biologically relevant questions that can be tackled. We make all the 380 probes described in this study —as well as additional probes which we are continuously adding to our database—available to the community as a non-profit

DNA FISH probe repository (http://ifish4u.org/browse), following the model of Addgene for plasmids. To our knowledge, this is the largest collection so far available of oligo-based DNA FISH probes targeting hundreds of loci all along the human genome, and as such it represents an invaluable resource for the research community. As we are continuously expanding our probe collection by including probes more densely spread all along the human genome, as well as probes targeting the genome of relevant model organisms, we anticipate that iFISH will greatly increase the number of investigators that routinely use DNA FISH in their research.

Another reason that, in our view, has prevented a larger community of researchers to utilize DNA FISH on a regular basis, is the fact that, despite the availability of oligo databases for designing DNA FISH probes, there is currently no freely accessible tool enabling users with little or no expertise in bioinformatics to quickly design probes in any number and pattern along a given genome of interest. Our freely accessible iFISH4U interface (http://ifish4u.org/probe-design) fills this gap in, by providing a user-friendly environment for designing both single probes, as well as sets of multiple probes, by simply tuning four intuitive design parameters (probe size, centrality, homogeneity and interprobe distance). Importantly, in addition to our in-house oligo databases, iFISH4U can accept as input any of the OligoMiner databases currently available (https://oligopaints.hms.harvard.edu/genome-files), as well as other custom-designed databases uploaded by the user, thus providing extreme flexibility.

Although enzymatic amplification of array-synthesized oligo-pools by sequential PCR, IVT, and reverse transcription is used in all the approaches previously described for making Oligopaints[3] and MERFISH[17] probes (summarized in Supplementary Data 1), here we have described a high-throughput pipeline that allows synthesizing hundreds of oligo probes in parallel, in 96- or 384-well plates, starting from a single synthetic oligo-pool. The turnaround time from PCR to ready-to-use probes is only 16 man-hours for producing 96 probes, each in three different colors, and even shorter by using liquid handling robotic devices. In this approach, the C adapter for probe detection is introduced during the PCR step. This means that the same oligo-pool can be used to generate multiple probes against the same target(s), each with a different C adapter, thus providing enormous flexibility in the number of possible probe combinations that can be generated in the same experiment. Using this pipeline, large numbers of individual probes can be stocked for long term as frozen PCR products, and then readily used for IVT whenever a specific probe is needed. Accordingly, we make all the 380 probes in our repository available to the community as purified PCR products that can be easily converted to ready-to-use probes (http://ifish4u.org/browse). This will enable individual researchers to rapidly produce large amounts of probes serving for multiple hybridization reactions in a highly cost-effective manner, without the need to design and purchase oligo-pools (see Supplementary Notes 1 and 2).

Thanks to our repository of probes targeting all the human autosomes and chrX, we have been able to explore the extent of intermingling between heterologous chromosome territories in human interphase cells, using a novel quantitative approach. Our chromosome-spotting approach is inspired by land surveying and uses multiple probes evenly spread along a given chromosome to visualize its territory as a cloud of fluorescence spots[2]. As we demonstrated here, chromosome-spotting probes are particularly suited for assessing intermingling, as the dotted nature of the signal enables the adoption of robust intermingling metrics, such as the mixing index that we have introduced. Our results indicate that mixing between heterologous chromosomes occurs at a seizable frequency in most of the cells, although the average mixing frequency is higher for certain chromosome pairs. These findings are in line with a recent study in Drosophila, which showed that most of the cells examined had some degree of overlap between heterologous chromosomes visualized using Oligopaints[4]. Our data also reveal that the frequency of inter-mingling is significantly higher in hESCs compared to fibroblasts, which might reflect the more accessible chromatin associated with pluripotency.

In conclusion, we have established a comprehensive and freely accessible resource that can be used to quickly design and produce large quantities of DNA FISH probes against multiple genomic regions of interest, as well as rapidly access a large panel of tested probes through our repository. We anticipate that iFISH will greatly facilitate the use of DNA FISH in a broad range of research and diagnostic applications.

## Methods

**Creation of the iFISH4U web interface**. We developed the iFISH4U probe design web interface (http://ifish4u.org/probe-design/) with a strong focus on versatility, as a Python3 package named `ifpd`. The interface allows a user to run one of two types of query: 'single query' or 'spotting query'. A single query (`ipfd_query_probe`) produces a list of potential single-locus probe candidates, each comprising $N$ oligos, in a given genomic region of interest. To achieve this, it first identifies all possible sets (herein, referred to as probe candidates) of $N$ oligos in the region of interest. Then, for each probe candidate, the interface computes three features: (i) size, (ii) centrality, and (iii) homogeneity. We defined the probe size as the difference between the genomic coordinates of the first base covered by its first oligo, and the last base covered by its last oligo. We defined the centrality $C_p(X)$ of a probe $X$ as 1 − the distance of the probe midpoint (in bp) and the midpoint (in bp) of the region of interest divided by half the size of the region of interest.

$$C_p(X) = 1 - \frac{\overline{MN}}{S/2} \tag{1}$$

Where $M$ is the probe midpoint, $N$ is the midpoint of the region of interest, $\overline{MN}$ indicates the distance between $M$ and $N$, and $S$ is the size of the region of interest. Centrality values span from 0, for a probe with the midpoint laying at the borders of the region of interest, to 1, for a perfectly central probe. We defined the probe homogeneity $H_p(X)$ of a probe $X$ as the reciprocal of the standard deviation of the distance between consecutive oligos in the probe candidate.

$$H_p(X) = \frac{1}{\sigma_d(X)} \tag{2}$$

$$\sigma_d(X) = \frac{1}{N-1} \sqrt{\sum_{i=1}^{N-1} \left(d_i(X) - \mu_d(X)\right)^2} \tag{3}$$

$$\mu_d(X) = \frac{1}{N-1} \sum_{i=1}^{N-1} d_i(X) \tag{4}$$

Where $N$ is the number of oligos in $X$, and $d_i(X)$ is the distance between the last base covered by the $i$-th oligo and the first base covered by the $(i+1)$-th oligo of probe $X$.

The user can rank these three features to optimize the probe design in an application-driven manner. The script uses the first ranked feature to select probe candidates in a narrow range (customizable by the user) around the best value of the first feature. Best values are defined as the minimum size, or maximum centrality, or maximum homogeneity across all the probe candidates. Then, the selected probe candidates are ranked based on the second feature, from the best to

the worst value. Finally, either all, or only the top $M$ candidates are reported, with $M$ being defined by the user.

The spotting query (`ifpd_query_set`) can be used to design a number $N$ of probes in a region of interest. This is achieved by initially dividing the genomic region of interest in $N+1$ windows $w$ of equal size. The single probe algorithm is then run on the first $N$ windows to identify an optimal probe candidate in each window, and the resulting probes are then collected to produce a candidate probe set. Then, the windows are shifted of a fraction of the $w$ size, and the algorithm iterates until the whole region of interest has been covered. Finally, homogeneity of inter-probe distance and probe size is calculated to rank the candidate probe sets and identify the optimal one. Specifically, we define the homogeneity of inter-probe distances and probe sizes of a probe set $Y$ as the reciprocal of the average between two values: the standard deviation of probe sizes, and the standard deviation of inter-probe distances.

$$H_s(Y) = \frac{2}{\sigma_D(Y) + \sigma_S(Y)} \tag{5}$$

$$\sigma_D(Y) = \frac{1}{M-1} \sqrt{\sum_{i=1}^{M-1} \left(D_i(Y) - \mu_D(Y)\right)^2} \tag{6}$$

$$\mu_D(Y) = \frac{1}{M-1} \sum_{i=1}^{M-1} D_i(Y) \tag{7}$$

Where $M$ is the number of probes in $Y$, and $D_i(Y)$ is the distance between the last base covered by the last oligo of the $i$-th probe and the first base covered by the first oligo of the $(i+1)$-th probe of the probe set $Y$.

We developed the interface to allow a user to easily compare multiple probe and probe set candidates (both in a tabular and graphical manner), to further explore single probe candidates and their oligo distribution, and to download them as either fasta, bed, or zip files. Furthermore, we implemented the `ifpd_mkdb` script to convert already available oligonucleotide databases (e.g., databases generated by OligoMiner) into a format that is immediately compatible with the interface. When the script re-formats a database, sequences can either be immediately provided as input to the interface, or automatically retrieved by querying the UCSC DAS server. Furthermore, we provide the `ifpd_dbchk` script to validate the generated databases for compatibility with our interface. The `ifpd` package can be promptly installed on a local machine serving Python3 and pip3, and the web interface can be run through the `ifpd_serve` script. Back-end scripts are available at http://github.com/ggirelli/iFISH-probe-design. The additional pages of the iFISH4U website were added as an ad hoc external costumization of the web interface provided in `ifpd` (http://github.com/ggirelli/iFISH4U).

**Creation of the human 40-mers database**. We extracted all the 40 nt sequences appearing only once in the human reference genome (Grch37/hg19 GCA_000001405.1) using `JELLYFISH` v2.2.6[35] and a custom-made pipeline in Perl which can be provided upon request. Afterward, we discarded the 40-mers with a homopolymer stretch of 7 or more bases, or with a GC-content outside of the 35–80% interval, and we calculated the average melting temperature of the remaining 40-mers. Subsequently, we filtered out the sequences with homology of 70% or higher to more than one genomic location using `VMATCH` v2.2.4. Afterward, we calculated the delta free energy of the most stable secondary structure at 65 °C using `OligoArrayAux` v3.8[36] and discarded the 40-mers with a negative value. We retained for further analysis only the 40-mers with a melting temperature in a range of 20 °C around the previously calculated average temperature. Lastly, we discarded overlapping 40-mers by starting from the first one and iterating through. We stored all the retained 40-mers in a sqlite3 database for easy access. To test for the presence of off-targets that might affect quality and efficiency of the probe hybridization, we used `Bowtie` v1.2.2 to align the 40-mers to the human genome reference, allowing up to 6 mismatches with the following command: *bowtie -f–seedlen 10–seedmms 3–maqerr 200 -k 1000–sam-mm hg19.gen-ome.faunique_oligonucleotides_2016_run40mer_h70.fasta.*

The `seedmms`, `seedlen` and `maqerr` parameters specify the seed length, the number of maximum mismatches allowed in the seed sequence and the threshold for the total number of mismatches in the final alignment, respectively. We used `sambamba` v0.6.7 to sort and index the alignment file, and a custom Python script to post-process and to collect statistics for each entry in our initial dataset. Briefly, for each oligonucleotide sequence, we search for the $NM{:}i < N >$ flag, which represents the minimal number of one-nucleotide edits (substitutions, insertions and deletions) needed to transform the oligonucleotide string into the reference string, and count the occurrences of each edit distance, from 0 to 6. Lastly, we flagged the 40-mers in the database that have more than 10 off-targets with 1–5 mismatches. Both the original and the flagged databases can be freely downloaded at http://ifish4u.org/download, and used for probe design through the iFISH4U interface.

**Comparison between iFISH and OligoMiner databases**. We downloaded all the OM hg19 databases ('Balance', 'Coverage', and 'Stringent') from https://oligopaints.hms.harvard.edu/genome-files. We divided the hg19 reference genome into non-

overlapping bins of different size, and calculated: (i) the percentage of bins with at last 96 oligos; (ii) the standard deviation of the distance between consecutive oligos in each bin. For the latter, we included all the oligos that overlapped with a bin (even if only partially). For each oligo database, we also calculated the genome coverage, as the fraction of the human reference genome that is covered by the oligos in the database. To compute the number of bins with at least 96 oligos, as well as the fraction of genome covered, we used the `rtracklayer` R package. To calculate the inter-oligo distance in each genomic bin, we used a custom-made Python3 script.

**Design of orthogonal and non-cross-hybridizing 20-mers**. We took advantage of a previously published list of 240,000 25-mers orthogonal to the human genome[23], and extracted from them all the 1,140,000 possible 20-mer substrings, using a custom-made pipeline (OOD-FISH v0.0.2) that can be made available upon request. We aligned the 20-mers to the human reference genome (Grch37/hg19 GCA_000001405.1) using `BLAT v36x1`[37] with the following parameters: `tileSize = 6 -stepSize = 1 -minMatch = 1 -oneOff = 1 -minScore = 0 -minIdentity = 0 -maxGap = 0 -repMatch = 131071 -noHead`. We discarded all the 20-mers with at least one alignment with maximum homology (defined as the fraction of single-base matches over the sequence length, i.e., 20 nt) equal to or higher than 80% of the oligonucleotide length (i.e., 16 nt). We calculated the self-dimerization free energy of all the 20-mers, and filtered out those with self-dimerization free energy ≤−5 kcal/mol. Then, we calculated the lowest heterodimerization free energy for every pair of 20-mers and their reverse complement sequence. We identified the largest set of 20-mers with hetero-dimerization free energy ≥−9 kcal/mol using the `Parallel Maximum Clique library`[38]. We calculated both self- and hetero-dimerization free energy values using the nearest-neighbor method for the longest matches stretch, assuming an oligo concentration equal to 0.25 M and a sodium concentration equal to 50 mM. All scripts used to design the orthogonal 20-mers are available at https://github.com/ggirelli/ood-fish/releases/tag/v0.0.2.

**Probe design**. Each probe is composed of 96 different oligos, each consisting of four parts (5′–3′): C (20 nt), F (20 nt), T (40 nt), and R (20 nt). To design the 330 probes uniformly spread along all human autosomes and chrX, we used the iFISH4U web interface (http://ifish4u.org), and selected T sequences in a way to minimize the probe size and homogeneity. We required that all the probes on the same chromosome are located as equidistantly as possible, by minimizing first the size of individual probes, then minimizing the probe homogeneity, and finally minimizing the inter-probe distance. Finally, we appended a unique combination of F and R adapters to each of the 96 T sequences in a probe, using a custom-made script in MATLAB. We adopted the same approach to design 46 extra probes on chr17, which were used to prepare the chr17 'dense' spotting probe. To design probes for comparing our oligo database with OligoMiner, we selected the oligos in the OM 'Balance' hg19 database that fall within the genomic region targeted by probe 2.22 on chr2, 5.16 on chr5, and 8.13 on chr8. For each probe, if less than 96 oligos were present in the OM 'Balance' hg19, we expanded the region of interest to the smallest larger region containing 96 oligos. To expand the regions in an optimal manner, we kept expanding each region to the closest oligo (either towards the 5-end or the 3′-end) until 96 oligos were found.

**Annotation of oligos in the 40-mers database and 330 probes**. We obtained the comprehensive human gene annotation (release 19 GRCh37.p13) from GENCODE (https://www.gencodegenes.org). We imported the annotation file into R using the `makeTxDbFromGFF` function. We extracted the transcript and non-overlapping intronic sets of sequences using, respectively, `exonsBy` and a combination of `intronsByTranscript`, `reduce` and `setdiff` functions. We also generated an exon-intron junction set using the last 25 nt of the exon and first 25 nt of the intron (and vice versa for the intron-exon junctions). We then matched the reverse complement of each oligo present in our database or in the 330 probes, against the set of transcripts using the `vcountPDict` function. Finally, we assigned each oligo to one of the following five categories based on its overlap with transcriptomic features: (i) no overlaps ('None'); (ii) exons-only; ('Exonic'); (iii) introns-only ('Intronic'); (iv) exons and introns ('Both'); (v) exon-intron junctions ('Junction'). We performed all the analyses in R, using functions from the `GenomicFeatures`[39] and `data.table` packages (https://github.com/Rdatatable/data.table/wiki).

**High-throughput probe production**. We developed a pipeline for large-scale, parallelized production of oligonucleotide DNA FISH probes, by slightly modifying and scaling up the workflows previously described to synthesize Oligopaints[3] and MERFISH probes[17,21]. A step-by-step protocol is available in Protocol Exchange[24]. Large numbers of probes can be produced through this pipeline using 96- or 384-well plates by adjusting the volumes accordingly. Pipetting steps in 96- or 384-well plates can be performed either using multi-channel dispensing devises, or fully automatically using liquid handling robots. In this work, we used 96-well plates and multi-channel pipettes to synthesize all the 330 probes. To synthesize the 63 chromosome spotting probes on chr17, we used a liquid handling device (I-DOT One, Dispendix), which allows easy and cost-effective combinatorial dispensing of

reagents in the nanoliter range. We purchased the oligos corresponding to the probes of interest as 12 K oligo-pools from CustomArray Inc. Each oligo-pool contains up to 12,000 unique oligos with the configuration shown in Fig. 1e. In this manner, up to 125 probes, each composed of 96 oligos, can be generated from a single array. However, depending on the size of the loci to be visualized, a higher or lower number of oligos per probe can be used. Importantly, since the color of each probe is specified during the amplification of the oligos, as described below, a single array can be used to re-generate multiple probes, each in different colors, for multiple cycles and in an efficient manner. We diluted each oligo-pool and dispensed equal volumes of the dilution are across a variable number of 96-well plates and wells, depending on the number of probes and colors per probe. We then amplified the oligos in each well by real-time PCR using the SYBR Select Master Mix (Thermo Fisher Scientific, cat. no. 4472913). We designed PCR primers that would anneal to the F and R adapters specific for each oligo in the pool, and incorporate the C adapter and T7 promoter sequence, on the 5 side of the F and R adapters, respectively. We purchased all the primers from Integrated DNA Technologies (IDT) as standard desalted oligos. We purified the PCR product in each well separately with Agencourt AMPure XP beads (Beckman Coulter, cat. no. A63881), and measured the DNA concentration using the Qubit dsDNA HS Assay Kit (Thermo Fisher Scientific, cat. no. Q32854). The PCR products generated in this way can be conveniently stored at −20 °C for several months, and used to quickly re-generate the probes when they are finished. Next, we converted each individual PCR product into RNA using the HiScribe T7 Quick high yield RNA synthesis kit (NEB, cat. no. E2040S). Each reaction was carried out at 37 °C for 12–16 h, in a final volume of 30 μL containing 1 μg of purified PCR product (either a single probe or a pool of probes), 6.67 mM of dNTPs, 2 Units of RNaseOUT Recombinant Ribonuclease Inhibitor (Thermo Fisher Scientific, cat. no. 10777019) and 2 μL of T7 RNA polymerase mix. We purified the amplified RNA (aRNA) with Agencourt RNAClean XP beads (Beckman Coulter, cat. no. A63987) and measured the RNA concentration with the Qubit RNA BR assay (Thermo Fisher Scientific, cat. no. Q10210). We then converted the purified RNA into cDNA by reverse transcription (RT) using the Maxima H Minus Reverse Transcriptase (Thermo Fisher Scientific, cat. no. EP0751) and a primer with the C adapter sequence (RT primer). Each reaction was carried out at 50 °C for 1 h, in a volume of 20 μL containing 15 μg of purified RNA, 1.5 mM of dNTPs, 20 μM of the corresponding primer, 1x reverse transcription buffer, 10 Units of Maxima H reverse transcriptase and 2 Units of RNaseOUT. We inactivated the enzymes by incubating the sample at 85 °C for 5 min. To remove the template RNA, we added 20 μL of 0.5 M EDTA and 20 μL of 1 M NaOH directly into the RT reaction, incubated the sample at 95 ° C for 15 min and immediately purified the single-stranded DNA (ssDNA) using Oligo binding buffer (Zymo Research, cat. no. D4060–1–40) and Zymo-Spin IC columns (Zymo Research, cat. no. C1004). We eluted each probe in 40 μL nuclease-free water and measured its concentration with the Qubit ssDNA Assay Kit (Thermo Fisher Scientific, cat. no. Q10212). We confirmed that the probes had the expected length by running them on Novex TBE-Urea Gels, 15% (Thermo Fisher Scientific, cat. no. EC6885BOX). We stored the probes at −20 °C prior to DNA FISH experiments.

**Probe complexity assessment by real-time PCR**. For each oligo in each selected probe, we designed a unique forward primer as the reverse complement of the first 20 nucleotides of the T sequence. As reverse primer for all the oligos within the same probe, we used the RT primer corresponding to the specific C adapter of each probe. As template for the PCR reaction, we either used individually amplified (probe 8.13 on chr8 and 18.3 on chr18) or a pool of probes synthesized in the same IVT reaction, including the probe of interest (a mix of 5 probes for probe 5.16 on chr5, and all the probes on chr2 for probe 2.22 on chr2). We purchased all the primers from Integrated DNA Technologies (IDT) as standard desalted oligos. We performed PCR reactions as following: first we prepared 12.5 μM dilutions of each primer, and 10 nM dilutions of each template. For each oligo, we mixed 3 μL of forward and 3 μL of reverse primers with 2 μL of the corresponding template. We then amplified each oligo individually using the SYBR Select Master Mix (Thermo Fisher Scientific, cat. no. 4472913) with the following protocol: 95 °C for 2 min followed by 40 cycles of 95 °C for 15 s and 60 °C for 60 s. We performed each reaction in triplicate in three wells of the same 384-well plate, together with a negative control reaction using water as template, for every oligo.

**Preparation of cells**. We purchased IMR90 fetal lung fibroblasts, A549 lung carcinoma cells, and HME human mammary epithelial cells from ATCC (cat. no. CCL-186, CCL-185, and PCS-600–010, respectively), and HAP1 chronic myeloid leukemia cells from Horizon Discovery (cat. no. C859). Human embryonic stem cells (HS975[40]) were derived and used following written consent of the donor and approval from Regional Ethical Review Board in Stockholm (2011/745–31/3). None of these cell lines is included in the ICLAC database of commonly mis-identified cell lines. We regularly checked all the cell lines for mycoplasma contamination, but did not authenticate them. We cultured the cells in 6-well chambered coverslips (custom-made by Grace Bio-Labs) or in coverslips immobilized onto a silicon gasket (CultureWell Multislip Cell Culture System MSI-12, Thermo Fisher Scientific, cat. no. C24760) in the following media: IMR90 cells in Minimum Essential Medium (MEM, Merck, cat. no. M4655) supplemented with 10% FBS, 1% non-essential amino acids (Thermo Fisher Scientific, cat. no.

11140035) and 1% L-glutamine (Thermo Fisher Scientific, cat. no. 25030081); A549 cells in Ham's F-12K (Kaighn's) Medium (Thermo Fisher Scientific, cat. no. 21127022) supplemented with 10% FBS; HAP1 cells in Iscove's Modified Dulbecco's Medium (Sigma-Aldrich, cat. no. I2911) supplemented with 10% FBS; HME cells in Medium 171 (Thermo Fisher Scientific, cat. no. M171500) supplemented with Mammary Epithelial Growth Supplement (Thermo Fisher Scientific, cat. no. S0155); primed hESCs in NutriStem hPSC XF Medium containing bFGF and TGFβ (Biological industries, cat. no. 05–100–1 A) on coverslips precoated with 10 μg/ml Human recombinant laminin-521 (BioLamina, cat. no. LN521–03) following the manufacturer's instructions. We incubated all cells at 37 °C in 5% O₂ and 5% CO₂. When the cells reached 80% confluency in each well (60% in case of hESCs), we processed the samples following an adapted version of the protocol for 3D-FISH that we recently described[25]. Briefly, we fixed the cells in 1x PBS (Thermo Fisher Scientific, cat. no. AM9625)/4% formaldehyde (EMS, cat. no. 15710) for 10 min at room temperature (RT), followed by quenching of unreacted formaldehyde in 1x PBS/125 mM glycine for 5 min at RT. Subsequently, we washed the cells three times, 5 min each with 1x PBS/0.05% Triton X-100 at RT and permeabilized them in 1x PBS/0.5% Triton X-100 for 20 min at RT. Following overnight incubation in 1x PBS/20% glycerol at RT, we subjected the cells to five cycles of freeze-and-thaw in liquid nitrogen (30 s in liquid nitrogen, thawing in ambient air, 2–3 min in 1x PBS/20% glycerol at RT), and then washed them three times, 1 min each in 1x PBS/0.05% Triton X-100 at RT. Afterward, we incubated the cells in 0.1 N HCl for 5 min and quickly rinsed them twice in 1x PBS/0.05% Triton X-100 at RT. Lastly, we rinsed the cells in 2x SSC buffer (Thermo Fisher Scientific, cat. no. AM9763) and incubated them overnight in 2x SSC/50% formamide/50 mM sodium phosphate at RT. The following day, we transferred the cells at +4 °C and kept them for one week in 2x SSC/50% formamide/50 mM sodium phosphate. For hESCs, we additionally incubated the cells in 2x SSC/50% formamide/50 mM sodium phosphate/0.1% Tween20 for 24 h at +4 °C in order to reduce autofluorescence in the AlexaFluor 594 channel. Lastly, we exchanged the buffer to 2x SSC at +4 °C and stored the cells in it up to 1 month.

**3D DNA FISH**. For probes targeting a single locus, we first incubated the samples for 1 h at 37 °C in a humidity chamber immersed in pre-hybridization buffer (PHB) containing 2x SSC/5x Denhardt's solution/50 mM sodium phosphate buffer/1 mM EDTA/100 ng/μL ssDNA/50% formamide, pH 7.5–8.0. During this time, we prepared the first hybridization mix (HM-1) by mixing the single-locus probes (up to six single-locus probes together; typically 1 pmol of each probe is used) at 1:9 v/v ratio with 1.1x first hybridization buffer (HB-1) containing 2.2x SSC/5.5x Denhardt's solution/55 mM sodium phosphate buffer/1.1 mM EDTA/111 ng/μL ssDNA/55% formamide/11% dextran sulfate, pH 7.5–8.0. We removed the coverslip from PHB and dispensed the sample on top of 10 μL of HM-1 a microscope slide. We then sealed the coverslip with fixogum (MP Biomedical, cat. no. 11FIXO0125) and waited until the fixogum became solidified. Then we performed DNA denaturation for 3 min at 75 °C on a heating block. Afterwards, we incubated the sample for 15–18 h at 37 °C. The next day, we washed the coverslip three times, 10 min each at 37 °C in 2x SSC/0.2% Tween, while shaking, followed by two washes, 7 min each at 58 °C in 0.2x SSC/0.2% Tween pre-warmed at 58 °C, inside a water bath, a brief wash in 4x SSC/0.2% Tween at RT, two brief washes in 2x SSC and one final short wash in 2x SSC/25% formamide. Next, we immersed the coverslip on 100 μL of the second hybridization mix (HM-2) containing the secondary fluorescently labeled oligonucleotides (one color per locus, up to six colors together), each at a final 20 nM concentration in 2xSSC/25% formamide/10% Dextran sulfate/1 mg/mL E.coli tRNA/0.02% bovine serum albumin/10 mM Vanadyl-ribonucleoside complex, and incubated the sample for 3 h at 30 °C. Afterwards, we washed the coverslip for 1 h at 30 °C in 2x SSC/25% formamide, followed by 30 min at 30 °C in 1.23 ng/mL Hoechst 33342 in 2x SSC/25% formamide. Lastly, we briefly rinsed the coverslip twice in 2x SSC, before mounting the sample with Prolong™ Diamond Antifade Mountant (Thermo Fisher, cat. no. P36965).

For chromosome spotting probes, we first subjected the samples to two consecutive pre-hybridization steps. In the first step, we dispensed a drop of Image-iT FX Signal Enhancer (Thermo Fisher Scientific, cat. no. I36933) into each well of the 6-well chambered coverslip, and incubated the cells for 1 h at RT. In the second pre-hybridization step, we removed the signal enhancer from each well and dispensed PHB containing 2x SSC/5x Denhardt's solution/50 mM sodium phosphate buffer/1 mM EDTA/100 ng/μL ssDNA/50% formamide, pH 7.5–8.0 in each well. We then incubated the sample for 1 h at 37 °C in a humidity chamber. During this time, we prepared the first hybridization mix (HM-1) by mixing the spotting probes (up to 6 chromosomes together, with each probe at a concentration of 6 nM) at 1:9 vol./vol. ratio with 1.1x first hybridization buffer (HB-1) containing 2.2x SSC/5.5x Denhardt's solution/55 mM sodium phosphate buffer/1.1 mM EDTA/111 ng/μL ssDNA/55% formamide/11% dextran sulfate, pH 7.5–8.0. We removed the PHB from each well and dispensed HM-1 in each well. We then sealed the 6-well chambered coverslip with PCR foil and performed DNA denaturation for ~3 min at 75 °C on a heating block. Afterwards, we incubated the sample for 15–18 h at 37 °C. The next day, we washed the coverslip three times, 10 min each at 37 °C in 2x SSC/0.2% Tween, while shaking, followed by two washes, 7 min each at 58 °C in 0.2x SSC/0.2% Tween pre-warmed at 58 °C, inside a water bath, a brief wash in 4x SSC/0.2% Tween at RT, two brief washes in 2x SSC and one

final short wash in 2x SSC/25% formamide. Next, we added to each well the second hybridization mix (HM-2) containing the secondary fluorescently labeled oligonucleotides (one color per chromosome, up to 6 colors together), each at a final 20 nM concentration in 2x SSC/25% formamide/10% Dextran sulfate/1 mg/mL E.coli tRNA/0.02% bovine serum albumin/10 mM Vanadyl-ribonucleoside complex, and incubated the sample for 3 h at 30 °C. Afterward we washed the coverslip for 1 h at 30 °C in 2x SSC/25% formamide, followed by 30 min at 30 °C in 1.23 ng/mL Hoechst 33342 in 2x SSC/25% formamide. Lastly, we briefly rinsed the coverslip twice in 2x SSC, followed by a short wash in 2x SSC/10 mM Tris-HCl/0.4% Glucose, before mounting the sample with 2x SSC/10 mM TROLOX/37 ng/μL Glucose Oxidase/32 mM Catalase. In the case of hESCs, the sample was mounted using Prolong™ Diamond Antifade Mountant (Thermo Fisher, cat. no. P36965).

**Comparison of iFISH and OMB probes**. To compare iFISH and OMB probes using similar hybridization conditions to those previously used for probes designed with OligoMiner[10], we applied the 3D DNA FISH procedure described above with the following modifications: after fixation, we washed the coverslips once with 2x SSCT/50% formamide for 2 min, followed by a wash with 2x SSCT/50% formamide for 20 min at 60 °C. We then air-dried the coverslips. For the first hybridization, we used a buffer containing 2x SSCT/50% formamide/10% dextran sulfate/0.4 μg/μl RNase A. We incubated the coverslips for 7 min at 37 °C and then performed denaturation for 3 min at 80 °C. We incubated the samples for 72 h at 47 °C in a humidity chamber. After the first hybridization, we washed the coverslips four times with 2x SSCT for 5 min at 60 °C, once with 2x SSCT for 5 min at room temperature, quickly rinsed them with 1x PBS, and finally air-dried them. The second hybridization and washes were performed as described for 3D DNA FISH above.

**Simultaneous DNA-RNA FISH**. We prepared the samples in the same manner as described for DNA FISH above, with few modifications. In order to preserve RNA, we added RNases inhibitors in all the buffers needed, following fixation in 1x PBS/4% formaldehyde. We heated all the buffers except for HCl at 60 °C for 10 min after adding RNAsecure RNase Inactivation Reagent (ThermoFisher, cat. no. AM7006) at 1:25 v/v ratio and Ribonucleoside Vanadyl Complex (RVC, New England Biolabs, cat. no. S1420S) at a final concentration of 10 mM. We dispensed pre-hybridization buffer (PHB) containing 2x SSC/5x Denhardt's solution/50 mM sodium phosphate buffer/1 mM EDTA/ 100 ng/μL ssDNA/50% formamide/10 mM RVC, pH 7.5–8.0 in each well. We then incubated the sample for 1 h at 37 °C in a humidity chamber. During this time, we prepared the first hybridization mix (HM-1) by mixing the small MYC probe at 1:9 v/v ratio with 1.1x first hybridization buffer (HB-1) containing 2.2x SSC/5.5x Denhardt's solution/55 mM sodium phosphate buffer/1.1 mM EDTA/111 ng/μL ssDNA/55% formamide/11% dextran sulfate/10 mM RVC, pH 7.5–8.0. We removed the PHB from each well and filled the wells with the HM-1. We then sealed the 6-well chambered coverslip with PCR foil and performed DNA denaturation for 3 min at 75 °C on a heating block. Afterward we incubated the sample for 15–18 h at 37 °C. The next day, we washed the coverslip three times, 10 min each at 37 °C in 2x SSC/0.2% Tween/10 mM RVC, while shaking, followed by two washes, 7 min each at 58 °C in 0.2x SSC/0.2% Tween/10 mM RVC pre-warmed at 58 °C, inside a water bath, a brief wash in 4x SSC/0.2% Tween/10 mM RVC at RT, two brief washes in 2x SSC/10 mM RVC and one final short wash in 2x SSC/25% formamide/10 mM RVC. Next, we added to each well the second hybridization mix (HM-2) containing the secondary fluorescently labeled oligonucleotides at a final 20 nM concentration and MYC smFISH probe at a final concentration of 3 ng/μL in 2x SSC/25% formamide/10% Dextran sulfate/1 mg/mL E.coli tRNA/0.02% bovine serum albumin/10 mM RVC, and incubated the sample for 3 h at 30 °C. Afterwards we washed the coverslip for 1 h at 30 °C in 2x SSC/25% formamide, followed by 30 min at 30 °C in 1.23 ng/mL Hoechst 33342 in 2x SSC/25% formamide. Lastly, we briefly rinsed the coverslip twice in 2x SSC before mounting the sample with Prolong Diamond Antifade Mountant (Thermo Fisher, cat. no. P36965).

**Image acquisition and image processing**. We imaged all the samples using a ×100 1.45 NA objective mounted on a custom-built Eclipse Ti-E inverted microscope system (Nikon) controlled by the NIS Elements software (Nikon) and equipped with an iXON Ultra 888 ECCD camera (Andor Technology). We acquired multiple image stacks per sample, each consisting of 49–60 focal planes spaced 0.3 μm apart. We converted raw images from ND2 (Nikon) or CZI (Carl Zeiss) format to uncompressed TIFF format using the nd2_to_tiff and czi_to_tiff scripts, respectively, from the our custom-developed pygpseq Python3 package available on GitHub: https://github.com/ggirelli/pygpseq/. We removed out-of-focus images by using the tiff_findoof v0.3.1 tool from the pygpseq package. The script identifies and discards stacks in which the peak of the gradient magnitude of the stack intensity over Z does not fall in a range of 50% of the stack around the mid slice. To correct for chromatic aberrations and shifts between channels, we imaged TetraSpeck Microspheres (0.1 μm, fluorescent blue/green/orange and dark red, Thermo Fisher Scientific, cat. no. T7279) before or after each imaging session. We used the DNA stain channel as the reference channel, and determined the location of the beads by fitting a 2D Gaussian profile in (x,y) and a 1D Gaussian in z by optimizing a maximum-likelihood functional using the

Nelder–Mead method[41]. Signals corresponding to the same bead were identified by clustering the strongest dots detected in the DNA channel with the strongest dots in each other channel. After determining the shift between the channels, we detected outliers and discarded them. Finally, we fitted a second order 2D polynomial deformation to the (x,y) plane, by using the remaining point pairs. Along the z directions, we only corrected for shifts[42]. After chromatic aberration correction, we performed automated 3D segmentation of cell nuclei stained with Hoechst 33342. To this end, we first deconvolved the DNA staining channel using the `Huygens Professional v17.04` Software (Scientific Volume Imaging), with the following parameters: `CMLE algorithm`, `null background`, and `signal-to-noise ratio (SNR) of 7, in 50 iterations`. We estimated a theoretical point spread function using the same software, considering both our microscope setup and the optical configuration used to acquire the images. After deconvolution, we performed 3D segmentation of the nuclei in each field of view, using the `tiff_auto3dseg` script of the `pygpseq` package. The script combines with a logical AND operation two binary masks, generated with the `threshold_otsu` and `threshold_local` methods from the `scikit-image.filters` package[43]. Then, it discards objects touching the XY contour of the image, fills any holes in the masks, and performs a dilate-fill-erode operation. To identify putative G1-phase cells, we adapted a previously published approach[44] that selects nuclei based on the integral of DAPI intensity over the nuclear volume. Briefly, the algorithm first tries to fit a sum of Gaussians to both distributions of the integral of DNA staining intensity over the nuclear volume, and of the nuclear area in the z-projection of the segmented images. If the fitting fails, the algorithm fits a single Gaussian instead. Lastly, we labeled as G1 the nuclei falling in a range of ±3 standard deviations around the mean of the major fitted Gaussian in both distributions.

**FISH signal identification and analysis**. To identify FISH dots, we used our in-house suite `DOTTER`, written in MATLAB (MATLAB and Statistics Toolbox Release 2018a) and C99 with GSL (https://www.gnu.org/software/gsl/). To be able to analyze a large number of cells, we set out to identify a semi-automated dot-picking procedure that would yield as similar results to manual dot-picking as possible. We found that the best results were obtained, for a given experiment, by selecting the same intensity threshold for all the fields of view, so that the resulting distribution of dot counts per nucleus would resemble as much as possible the expected distribution based on the ploidy of the cells analyzed. It is important to note that this semi-automated dot picking procedure, while giving the most unbiased results and allowing to analyze thousands of cells in a reasonable amount of time, is more prone to a false-positives and false-negatives compared to a procedure in which a different threshold is manually set for each field of view. The first exception to this procedure is how we analyzed the 'OMBshort' probes datasets. Since those probes yielded no signal in most of the cells, and very dim putative signals were only sporadically seen, we manually selected a threshold for this dataset after a careful inspection of the images by eye. The second exception to the aforementioned rule is the way in which we compared the iFISH and the OMB probes, the results of which are shown in the Supplementary Fig. 2f–g. Here, we first identified the most suitable threshold for a given OMB probe by inspecting multiple fields of view by eye and manually selecting a threshold that seemed to best capture the true dots, and then by applying the same exact threshold to the images of the corresponding iFISH probe. To calculate the signal-to-noise ratio for a dot, we computed the ratio between the intensity of the dot over the local background, defined as the mean intensity at the edge of a square of size 11 × 11 pixels centered at each dot.

**DNA-RNA FISH image analysis**. To analyze the *MYC* DNA-RNA FISH results, we first visually inspected each cell individually and retained only cells with a single clearly visible transcription center (TC, corresponding to the site of *MYC* transcription) in order to simplify the analysis. We then identified both DNA and RNA FISH dots using our suite `DOTTER`, after correcting chromatic abberations as described above. We classified the manually selected TCs as true-positives, if they had a DNA FISH dot (corresponding to the *MYC* locus) in close proximity (≤800 nm in 3D), or otherwise as false-negatives. We calculated the sensitivity as $TP/(TP + FN)$, where $TP$ is the number of true-positives and $FN$ the number of false-negatives.

**Accessibility and expression of probe targets**. To test the possible effect of local DNA accessibility on the quality of the 330 probes, we first downloaded previously published HAP1 ATAC-seq data[45] from the GEO repository GSE111047 (SRR6766909, SRR6766910, SRR6766911), and processed them using the Parker Lab's ATAC-seq processing pipeline (https://github.com/ParkerLab/ATACseq-Snakemake). We used the `bedGraph` files generated by the pipeline to calculate the mean ATAC-seq signal inside each of our 330 probes (min probe size: 7099 bp; median: 7938 bp; max: 21,603 bp). As a control, we assessed the ATAC-seq signal in 10 kb genomic windows centered on 2045 non-overlapping locations, equally distributed across the genome. For analysing gene expression, we obtained HAP1 RNA-seq raw files from the GEO repository GSE95015. Before mapping the reads, we removed adapter sequences using `TrimGalore` v0.4.4_dev and we discarded

reads shorter than 16 nt. We first mapped the retained reads using `Bowtie2` v2.3.4.1 (parameters: `-sensitive-local`) to a list of human rRNAs and tRNAs, retrieved from NCBI (https://www.ncbi.nlm.nih.gov/). We then mapped the reads that failed to align to UCSC hg19/GRCh37 genome assembly using the Gencode (v27) gene annotation as reference and the `STAR` software v2.6.0c (parameters: `-twopassMode Basic -alignSJoverhangMin8 -alignSJDBoverhangMin 1 -sjdbScore 1 -alignIntronMin 20 -alignIntronMax 1000000 -alignMatesGapMax 100000 -outFilterMultimapNmax 1 -outFilterMismatchNmax 999 -outFilterMismatchNoverReadLmax 0.04`). We discarded PCR duplicates using the `Picard MarkDuplicates` v2.18.11 module. To quantify the RNA-seq signal at gene level, we used the `QoRTs QC` module v1.3.0 (parameters: `minMAPQ 255`) and Gencode (v27) gene annotation as reference. We normalized gene counts to TPM (Transcript Per Million tags). To quantify the RNA-seq signal at 1 Mb resolution, we used the `bamCount` function from the `bamsignals` package v1.12.1 in R. To assess if local DNA accessibility and gene expression levels influence the performance of iFISH probes, we grouped the 153 individually tested probes in three accessibility groups, based on their ATAC-seq signal (low: <33th percentile; medium: ≥33th percentile and <66th percentile; high: ≥66th percentile). Lastly, we computed the distribution of mean dot counts per cell and median signal-to-noise ratio in each group, using custom-made R scripts. We applied the same procedure to categorize the probes into three expression groups, based on their RNA-seq signal (TPM), and computed the distribution of mean dot counts per cell and median signal-to-noise ratio in each group, using custom-made R scripts.

**Estimation of chr17 volume**. We first identified FISH signals using `DOTTER`, setting for each channel a threshold based on the brightness of the dots, in order to get, on average, the expected number of dots per nuclei. We removed dots that were more than 1300 nm to the closest dot. For each nucleus, we clustered the dots from all the channels using 2-means clustering, except when the GAP statistics suggested there was only one cluster[46]. Then, for each cluster, we created all possible sub-clusters using one, two, three, or all four channels. From all the sub-clusters we selected those in which the number of dots was within ±30% of the expected number of dots in each channel. Lastly, we calculated the volume of the convex hull of each dot cluster.

**Intermingling analysis**. To assess the extent of intermingling between heterologous chromosomes we devised three different approaches. In all the cases, we retained for analysis chromosome pair instances in G1-phase cells in which the number of dots per chromosome per cell was comprised between ±30% of the number of probes of each chromosome times 2 (number of dots expected in G1 diploid cells). In the first approach, for each chromosome pair per cell, we defined the volume of the nucleus comprised between the minimum and maximum of the three Cartesian coordinates of all the dots. We then divided the volume into an arbitrary number of cubes (we tested cubes with side equal to 5, 10, and 15 pixels), and for each cell we checked whether it contained any number of dots from both chromosomes in each pair. Finally, we defined the mixing index as the percentage of cubes containing dots of both chromosomes over the total number of cubes with dots. In the second approach, for each chromosome pair per cell, and for each dot per chromosome, we checked whether the nearest neighbor dot belonged to the same (homologous nearest neighbor) or to the other chromosome in the pair (heterologous nearest neighbor). We then defined the nearest neighbor index as the mean percentage of the dots in one chromosome having a heterologous nearest neighbor in each cell. For each chrA-chrB pair, we performed the same analysis in both directions, i.e. we calculated the nearest neighbor index for chrA vs. B and for chrB vs. A. We performed all the analyses using custom scripts written in MATLAB.

**Dot clustering**. To assess how well FISH dots can be clustered into two separate territories, for 6 chromosomes visualized in both hESCs and IMR90 cells, we extracted the dot coordinates per cell. We retained for analysis only G1-phase cells in which the number of dots per chromosome per cell was comprised between ±20% of the number of probes for that chromosome times 2 (number of dots expected in G1 diploid cells). We then applied the `kmeans` and `silhouette` functions in MATLAB to group all the dots belonging to the same chromosome in a cell into two separate clusters and to compute the mean silhouette value per cell of the clustered dots.

**Reporting summary**. Further information on experimental design is available in the Nature Research Reporting Summary linked to this article.

## Data availability
Processed images and coordinates of identified dots can be provided upon request. The source data underlying all Figures and Supplementary Figs. (except Supplementary Fig. 4c–e) are provided as a Source Data file. The following publically available datasets were used:

(1) Human OligoMiner databases: https://oligopaints.hms.harvard.edu/genome-files
(2) Human reference genome (release Grch37/hg19): https://www.ncbi.nlm.nih.gov/assembly/GCF_000001405.13/
(3) Human gene annotation (release 19 GRCh37.p13): https://www.gencodegenes.org
(4) ATAC-seq data from HAP1 cells: https://www.ncbi.nlm.nih.gov/geo/query/acc.cgi?acc=GSE111047
(5) RNA-seq data from HAP1 cells: https://www.ncbi.nlm.nih.gov/geo/query/acc.cgi?acc=GSE95015

## Code availability

All the custom code used for designing the 40-mers database, the probe design pipeline, and the iFISH4U web interface is provided as Supplementary Software and is also available at the following GitHub links:
(1) Oligonucleotide database: https://github.com/BiCroLab/oligo-picker (https://doi.org/10.5281/zenodo.2565946)
(2) Probe design pipeline: https://github.com/ggirelli/ifish-probe-design (https://doi.org/10.5281/zenodo.2565789)
(3) iFISH4U website extension: https://github.com/ggirelli/iFISH4U (https://doi.org/10.5281/zenodo.2565961).

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

## Acknowledgements

We thank Lei Xu (Bienko lab) for help with probe production. We acknowledge Mihaela M. Martis at the National Bioinformatics Infrastructure Sweden for providing scripts and resources needed for the construction of the 40-mers database. This work was supported by grants from Sony Imaging Products & Solutions, Inc. to M.M.; by grants from the Swedish Research Council (2013–32485–100360–69), the Knut and Alice Wallenberg Foundation (2016.0121 & 2015.0096), the Ming Wai Lau Centre for Reparative Medicine

(2–343/2016), CIMED (2–388/2016–40), and the Ragnar Söderberg Foundation (M67/13) to F.L..; by grants from the Swedish Research Council (521–2014–2866), the Swedish Cancer Research Foundation (CAN 2015/585), and the Ragnar Söderberg Foundation to N.C.; and by grants from the Science for Life Laboratory, the Karolinska Institutet KID Funding Program, the Swedish Research Council (621–2014–5503), the Human Frontier Science Program (HFSP Career Development Award), the Ragnar Söderberg Foundation, and the European Research Council (StG-2016_GENOMIS_715727) to M.B.. E.W. was supported by a scholarship from the Svenska Sällskapet för Medicinsk Forskning (SSMF).

## Author contributions

Conceptualization: E.G., G.G., M.M., M.B., N.C. Data curation: G.G., E.W.; Formal analysis: G.G., N.C., F.A., E.G., E.W. Funding acquisition: M.M., N.C., M.B. Investigation: E.G., J.C., A.M., X.L. Methodology: M.M., G.G., E.G., M.S., K.F., X.L., J.C., J.P.S., F.L., N.C., M.B. Project administration: M.B., N.C. Resources: J.P.S., F.L., Multidisciplinary Center for Advanced Computational Science at Uppsala University (Milou cluster), and National Supercomputer Centre at Linköping University (Triolith cluster). Software: G.G., E.W. Supervision: M.B., N.C. Validation: E.G., J.C., A.M., M.S., R.M. Visualization: G.G., E.G., M.B., N.C. Writing: N.C., M.B, E.G., G.G.

## Additional information

**Competing interests:** The authors declare no competing interests.

