## [Peer Review File · Nature Communications]

Reviewers' Comments:

Reviewer #1:

Remarks to the Author:

This work showcases diverse applications of oligo-pool derived probes for imaging DNA and RNA in cells. The images are beautiful. The text is clearly explained. However, the authors identify as the motivation for their work, the problem in the "lack of free resources that allow rapid design and generation of large numbers of DNA FISH probes at an affordable cost". It was not clear to me precisely what the current work added that was not already available (discussed more below). "a genome-wide human database of optimally designed oligonucleotide sequences" is already freely available, as are collections of "orthogonal adapter sequences" (though it would be great to see a thorough, quantitative evaluation of the efficacy of these different adapters, but that is not provided here). The method used to turn these sequences into probes quite closely follows published protocols in general conception and specific detail, and it appears to me the few deviations from the published versions increase the cost and complexity and decrease the efficiency rather than the reverse (see below). No quantitative comparison between approaches is described nor any evidence of further optimization of parameters relative to prior schemes. Diverse and beautiful applications of these probes are shown in cultured human cell populations, though many of these applications have been previously demonstrated with oligo-pool derived probes synthesized with a method extremely similar in many respects. There does not appear to be a particular set of biological conclusions from these applications aside from their quality. The authors eliminate the need for computational oligo selection for applications on human cells, by producing a pre-assembled list of probes for the human genome. Previous work (e.g. Chen...Zhuang, Science 2015, Boettiger...Zhuang, Nature 2105) relied on a rather convoluted probe design and pipeline which included dependency on some difficult to obtain software from built for microarray design. However, more recent work has already addressed this gap. For example, similar pre-built probe lists for the human genome and genomes of other model systems are already freely available online from the Wu lab at HMS (<https://oligopaints.hms.harvard.edu>). Several open source software packages have been released since which remove the dependence on the less user-friendly software from earlier publications and provide relatively easy to use software for the construction of oligo-pools e.g. (<https://github.com/brianbeliveau/OligoMiner>, <http://zhuang.harvard.edu/merfish.html>) see Moffitt...Zhuang, Methods Enzymol 2016, PNAS 2016a, PNAS 2106b; Beliveau...Yin PNAS 2018).

The details of the probe design describe here are very similar to previously published approaches (e.g. Chen...Zhuang, Science 2015, Boettiger...Zhuang Nature 2016): 40 nt, non-overlapping, limited GC range 35-80%, homopolymer stretch of 5, similarity of 70% to other genomic regions removed, primers/adapters selected from the same previously published 25-mers and truncated to 20mers. I don't think this is a barrier to publication, but it would be nice to see results from a rigorous test of which of these parameters matter and how much they influence quantitative properties of the assay, rather than presenting a close application of previously parameters. Are the minor differences in these designs improving the staining quality (e.g. probe uniqueness being screened on a 70% similarity instead of rejecting off-target based on a Tm differential or a 16 continuous base match described in prior work, 40mers instead of 42mers etc.)? To argue these are distinct a head-to-head comparison should be presented, though it does not appear that the current manuscript is claiming per se to achieve superior labeling or to use a meaningfully distinct set of parameters. This seems to be a missed opportunity. I suspect multiple labs have collected some data on 30mers vs 40mers, or effect of limited GC-range, the effect of excluding homopolymers, but I am not aware that data is published nor that probe designs optimized for RNA FISH are optimal for DNA FISH and v.v. or whether the probe design and hybridization protocol if co-optimized would come to the same value (aside from the Xu...Elledge microarray binding screen of DNA oligos, which is not in vivo and may differ substantially from binding on cells in traditional FISH buffers).

The probe synthesis protocol does not deviate substantially from published protocols (e.g.

Chen...Zhuang, Science 2015), with the notable exception of a few added steps on purification and quantification of RNA that are not explained nor contrasted with prior methods. As these added steps take time, require mildly expensive reagents (magnetic beads, a sorting magnet, QuBit reagents and system) and result in some loss of sample material, it appears to me that the previously published protocol was cheaper, faster, easier and more efficient. Previous reports showed that RNA purification is dispensable in this protocol, and that the RT reaction can be run with saturating amounts of primer to ensure maximum conversion to ssDNA (removing the need for precise quantification prior to the RT step). Excess primer from the saturating reaction is removed in the oligo-cleanup. Given the emphasis on cost, ease of use, and speed, these modifications are confusing. Purification is likely to result in some loss of product. QuBit quantification is also a non-trivial cost in probe synthesis and sacrifices a small amount of product and time. If the authors' analysis has revealed an advantage of this additional step (which adds cost and complexity to the protocol) it would be nice to see that data presented in the manuscript.

Another minor difference with potentially negative impact on the protocol is the use of RNase OUT in place of RNasin Plus as the RNase inhibitor in the RT reaction. Is RNase OUT as stable as RNasin Plus at 50C? Many RNase inhibitors lose efficacy at 50C, which results in loss of product.

While it is valuable to confirm the efficiency of these methods outside of the lab and collaborator teams which developed them, there is clearly much room to explore which would substantially enhance the impact of the paper. For example: Does non-overlapping probes actually help? Would overlapping probes work better? (It is unlikely all the probes stick to their targets in all the cells, given the spot-detection rates for small batteries of probes). Do all these 'orthogonal primers' work equally well (quantitatively)? The authors own data as well as comments in the community would suggest not – 15 – 33% of probes did not even amplify sufficient for testing in the first round, let alone a quantitative comparison of how abundance, library complexity, and resulting contrast/brightness in FISH images. How many adapters were tested? Do all work equally well?

I had also a few technical concerns:

I would have liked to see controls of samples labeled with Myc-RNA only to help evaluate the efficiency of the RNA-DNA FISH protocol that is described. One might expect that the some of the RNA probe is washed out during the addition of the DNA probe and denaturation of the sample, which the authors perform after RNA probe hybridization. Evidently this does not entirely remove the signal from this highly expressed transcript (the authors show beautiful double-stained images), but it might reduce the efficiency and accuracy of the otherwise quantitative smFISH approach (e.g. the number of cytoplasmic spots detected might be reduced).

It seems the DNA FISH probes will also label nuclear RNAs, unless the probes are explicitly designed to be sense to each gene or the sample is treated with RNase (which is not feasible in the combined experiment). This cross-talk may not matter in all applications, but should at least be commented on.

The description of the chromatic correction protocol is incomplete, which is particularly problematic for a methods-focused manuscript. The authors report they imaged tetraspec beads before and after imaging cells, but do not describe how they used this data to perform corrections.

Presumably they computed the chromatic displacements between all beads, calculating a field correction by computing a 2D polynomial (3rd order? 4th order?) map which corrects these aberrations, and applied this field correction to the corresponding data channels. A 3D map would be better but it cannot be computed from beads restricted to the 2D plane of the coverglass. For this the authors would have done better to affix the beads to the cells, which would distribute them throughout the 3D volume imaged and allow for a 3D correction map to be computed (as is typical for sub-diffraction limit chromatic corrections, see Sigal...Zhuang 2015 or Boettiger...Zhuang 2016 for recent examples). With a Plan-Apo chromatically correcting objective these aberrations should be small (largely <200 nm, increasing radially from the center of objective) and may not be a major concern for larger scale features described in this work. However, the approaches used

should be clearly described.

For cell cycle identification -- the described method based on DAPI intensity sounds very approximate. It would be nice to validate the DAPI-based approach with some cell cycle markers. For example, the FUCCI system, or an antibody against one of the more abruptly-changing cell-cycle associated proteins, e.g. Geminin. As the cell cycle claims are not essential to the major claims of the manuscript and the methodology orthogonal to the IFISH presented improving the cell-cycle discrimination is not essential.

Reviewer #2:

Remarks to the Author:

iFISH is a free resource for large-scale parallelized design and production of DNA FISH probes, by E. Gelali, et al., submitted to Nature Communications

E. Gelali et al. describe 330 oligo-based probes for use in studies of human genome organization via fluorescent in situ hybridization (FISH). As proof-of-principle, they target MYC (1 Mb as well as ~14 kb), then 6 locations on chromosome 18, and, finally, 330 locations (using any of three dyes) spaced either ~10 or ~5 Mb apart across chromosomes X and 1-22. The authors then demonstrate the usefulness of these probes for defining chromosome territories (including the intersection of overlapping territories) as well as labeling mitotic chromosomes, discovering two intriguing patterns – that the two copies of chromosome 17 cluster more than would be expected, and that chromosome territories are less distinct in hESC cells than in IMR90 cells. The images are compelling, the data believable, and the writing accessible. Importantly, a 330-probe library will be of great help to laboratories that currently do not have the expertise or resources to design and/or produce probes on their own. What is unclear, however, is the magnitude of advance over published methods based on and/or related to the technology called Oligopaints. Specifically, it seems that probes produced by the described method, iFISH, are the same in structure as those produced by Oligopaints. As with iFISH, Oligopaints can also be multiplexed to produce multiple individual probes, amplified by T7 polymerase, indirectly labeled, and, if desired, labeled with different dyes (e.g., Beliveau et al. 2015 Nat Commun, Chen et al. 2015 Science, Murgha et al. 2015 Biotechniques, Boettiger et al. 2016 Nature, Shah et al. 2016 Neuron, Wang et al. 2016 Science, Cattoni et al. 2017, Eng et al. 2017 Nat Meth, Bintu et al. 2018 Science, Szabo et al. 2018 Sci Adv, etc.). There is also significant redundancy with published protocols (most recently, OligoMiner from Beliveau et al. 2018 Proc Nat Acad Sci USA) and databases (Oligopaints) for genomic targets. Very recently, Oligopaint probes have even been used to examine chromosome territories (Rosin et al. 2018 PLOS Genetics). Additional comments are as follows:

a) Page 2 (Abstract): "...there is little recognition of DNA FISH as a powerful tool that per se allows investigating unique aspects of chromosome organization in single cells that cannot be captured by Hi-C." This statement seems out of place, as many researchers, including those in the Hi-C world, would argue that DNA FISH has made significant contributions to what we know about genome organization.

b) Page 2: "...DNA FISH has lagged considerably behind 3C methods with respect to the number of research groups that are using it as a routine technique." This statement is puzzling, as it greatly understates the widespread recognition and use of DNA FISH.

c) Page 3: Prominent papers producing oligo-based probes that should be cited in the Introduction along with references 3 and 4 would be Navin et al. 2006 (Bioinformatics), Yamada et al. 2011 (Cytogenet Genome Res), and Boyle et al. 2011 (Chromosome Res).

d) Results: As noted in the summary statement, it is unclear how iFISH differs from Oligopaints in terms of structure and application. (See papers cited above.)

e) Results: As many of the probes correspond to <100 oligos, do the authors have information regarding the potential for skewed representation of the individual species of oligos after rounds of amplification?

f) Page 7: Please provide number of nuclei imaged (n) for Figure 1k.

g) Page 8: The authors are encouraged to incorporate mention and discussion of Rosin et al. 2018 (PLOS Genetics) into their Introduction and Discussion, respectively, as it examines the intermixing of chromosome territories in *Drosophila*.

h) Page 9: Regarding Figure 4c, might the authors provide an explanation in the text for the choice of which pairs of chromosomes were studied? Might the format of a matrix be a better way in which to present the results?

i) Page 10: "For more than two decades, DNA FISH has remained a highly specialized technique rarely used outside of clinical genetics laboratories. With the advent...interest in DNA FISH (page 10, Discussion)." As in comments a and c, above, I do not believe this statement accurately portrays the field.

j) Page 11: The Discussion contrasts iFISH to MERFISH, pointing out how, in iFISH, a) the library can be used to generate many individual probes, b) the identity of each probe is encoded in the adapter, and c) the dye for iFISH is introduced during amplification. Thus, authors are directed to Beliveau et al. 2012 (Proc Nat Acad Sci USA) and 2015 (Nat Commun) and Boettiger et al. 2016 (Nature) for comparable protocols in terms of the generation of individual probes from multiplexed libraries, use of barcodes (adaptors), and introduction of dyes during amplification.

Point-by-point response to the Reviewers' comments

Reviewer #1

This work showcases diverse applications of oligo-pool derived probes for imaging DNA and RNA in cells. The images are beautiful. The text is clearly explained.

We are grateful to this Reviewer for appreciating the quality of our images and strive for presenting our results as clearly as possible. We also thank the Reviewer for her/his constructive criticism and suggestions, which greatly helped us to improve our original manuscript, by re-focusing the key message which we want to convey. In our revised manuscript, we strive to make the major goal of our work as clear as possible, and that is to make DNA FISH a technique as user-friendly as possible. Therefore, we no longer emphasize the development of a database of oligos targeting the human genome, nor the pipeline which we implemented for high-throughput probe production. Instead, we now present iFISH as a powerful resource consisting of a large and continuously expanding repository of individually tested probes, and of web-based user-friendly tools for designing FISH probes.

To make our resource truly impactful, we have now created a comprehensive web interface, <http://ifish4u.org>, which enables two key functionalities that so far have not been implemented by other public tools:

- 1) Browsing and requesting probes in our growing repository, following the successful model of publicly available non-profit repositories of biological reagents, such as Addgene for plasmids.
- 2) Designing new probes (either single probes or large probe sets targeting multiple loci on one or more chromosomes), using a set of biologically intuitive parameters and a set of oligo databases, including those previously generated with OligoMiner.

We believe that these key features make iFISH an invaluable and, thus far, unrivaled resource for the whole 3D genome biology field, but also have a potentially far-reaching impact in other research fields, as well as in diagnostics. We hope that the Reviewer will appreciate our efforts towards enabling as many researchers as possible to benefit from the power of DNA FISH in their daily research, and strive to provide impactful high-quality tools following the spirit of open science.

Here we provide a point-by-point replies to the Reviewer's comments:

However, the authors identify as the motivation for their work, the problem in the "lack of free resources that allow rapid design and generation of large numbers of DNA FISH probes at an affordable cost". It was not clear to me precisely what the current work added that was not already available (discussed more below). "a genome-wide human database of optimally designed oligonucleotide sequences" is already freely available, as are collections of "orthogonal adapter sequences" (though it would be great to see a thorough, quantitative evaluation of the efficacy of these different adapters, but that is not provided here).

As stated above, while in our revised manuscript we do acknowledge that both oligo databases and databases of orthogonal adapter sequences already exist. At the same time, we also emphasize the fact that, as of today:

1. There exists only one ready-to-use set of oligo databases (OligoMiner), which, as shown in the **new Fig. 1c-d and Supplementary Fig. 1**, does not provide homogenous genome coverage, and is suboptimal especially for designing small-sized probes. To our knowledge, these databases have been quantitatively assessed mainly by designing probes consisting of thousands of oligos and targeting large (> 1 Mb) genomic regions.

2. Oligopaint probes, which represent the most commonly used oligo-based probes, have been commonly designed using the OligoArray software, which is not user-friendly. In fact, depending on the publication, different *ad hoc* strategies for designing Oligopaint probes have been described, which can easily confuse potential users (including ourselves). Our literature review summarized in the **new Supplementary Table 1** shows that there is very little homogeneity in the set of parameters and filters that were used to design oligos in Oligopaint and MERFISH probes.
3. There is no freely available tool that can be used by non-experts in bioinformatics to quickly and easily select oligos from the available OligoMiner databases to design FISH probes with pre-defined characteristics.
4. There exists a database of 25-mers orthogonal to the human genome (Xu et al, *PNAS* 2009; PMID: 19171886). Yet, the list of barcodes that have been reportedly used to create Oligopaints is very short (at most, 17 different 20-mer barcodes listed in Moffitt et al, *PNAS* 2016; PMID: 27625426), and, as such, would not allow for the large-scale probe production as the one we are describing here.

In this manuscript, we have addressed these issues using the following approach:

1. We have created a new database of 40 nt oligos (40mers), which shows much higher and much more homogenous coverage of the human genome compared to the available OligoMiner hg19 databases. We believe that the fact that the OligoMiner (OM) databases do not provide a homogenous genome coverage, as well as sufficient coverage to allow for probing of small genomic regions (<15 kb), on its own justifies the development of a new database. However, we are aware that the creation of a new database does not warrant its utility. Therefore, we individually tested and validated more than 150 probes designed using our database, to secure its usability. We present the results of this large-scale validation effort in the **new Figure 2a and 2b**.

For a few selected regions, we also compared probes designed using our new oligo database ('40mer probes') with probes designed using the OM 'Balance' hg19 database ('OM probes'). One might argue that, given the lower coverage of the OM databases, they may contain higher quality oligos (more specific and with higher affinity) and that, perhaps, higher-coverage databases contain oligos of lower quality (with respect to their performance in DNA FISH). As a consequence, one would require less OM oligos per probe to detect equally good signals. We designed our benchmarking strategy precisely with this argument in mind. We selected two small regions (~8 kb), which we could target with the optimal number of oligos (96, based on our experience with our database), and identified all the available oligos targeting these regions in the OM 'Balance' hg19 database (OMB). The two OMB probes contained 38 and 27 oligos, respectively, which is similar to the number of oligos per probe recently reported in a paper using Oligopaints designed with OM (Nir et al, *PLoS Genet* 2018; PMID: 30586358). We reasoned that, if the oligos in the OM databases were indeed superior to our 40mers, we would be able to detect FISH signals even with these probes. However, as shown in the **new Supplementary Fig. 2a-e**, this was not the case. In fact, even when we compared 40mer and OMB probes containing 96 oligos, 40mer probes yielded FISH signals of higher intensity, for all the three probes tested (in the case of OMB probes, we designed them by extending the size of the regions targeted by the corresponding 40mer probes until 96 oligos were found, to ensure that the probe would have the smallest possible size, see revised **Methods**).

We should add that the approach we used to compare iFISH and OMB probes, as shown in the **new Supplementary Fig. 2a-e**, is the most favorable for the OMB probes. In fact, when we applied a more stringent way of comparing the two types of probes, by choosing

the same intensity threshold for the matched probes, the OMB probes were clearly performing worse than our probes, not only in terms of FISH signal intensity, but also of dot counts (see plots below). However, given that our intention is to provide new useful tools for the community, we would prefer not to include these data in the manuscript. In fact, we think that for experimental benchmarking to be fair and credible, it should be conducted in the frame of a large cooperative group, rather than by a single lab. We believe that the systematic *in silico* comparison between oligo databases which we have done should be sufficient to justify our work.

Figure legend. Left: distributions of dot intensities, for the iFISH and OMB probes containing 96 oligos, described in the **new Supplementary Fig. 2a-e**, using the same DNA FISH protocol as for all the 153 individually tested probes described in this work, in HAP1 cells. Right: distributions of dot counts per cell, for the same probes shown on the left.

For the same reason, we have decided to show here, but not include in the revised manuscript, the results of another benchmarking experiment, in which we compared the two aforementioned sets of probes using the DNA FISH protocol used in the latest work based on OM-designed Oligopaints by the Wu group at Harvard (Nir et al, PLoS Genet 2018; PMID: 30586358). We reasoned that OMB probes might perform better than iFISH probes under the same experimental conditions that were used to validate them. However, as it can be seen in the plots below, even under these optimized conditions, OMB probes performed worse than the corresponding iFISH probes.

Figure legend. Left: distributions of dot intensities, for the iFISH and OMB probes containing 96 oligos, described in the **new Supplementary Fig. 2a-e**, using the same DNA FISH protocol as described in Nir et al (PLoS Genet 2018; PMID: 30586358), in HAP1 cells. Right: distributions of dot counts per cell, for the same probes shown on the left.

- Given the rather confusing lack of a consensus and of a standardized protocol for designing oligo-based DNA FISH probes, we are convinced that, to prevent researchers from turning against the use of this powerful technique, it would be valuable to provide the community with a well-characterized large repertoire of ready-to-use FISH probes. Hence,

we decided to establish an open-source repository of FISH probes, which we are now continuously expanding.

3. We developed a web interface, iFISH4U (<http://ifish4u.org/probe-design>), that enables intuitive FISH probe design. iFISH4U allows, for the first time, to tune the design of FISH probes based on four features: 1) probe size; 2) probe centrality; 3) probe homogeneity; and 4) inter-probe distance when multiple probes targeting different loci on the same chromosome are designed. Depending on the application, probe size may range from a few kilobases to megabases, where small probes allow for high-precision localization of FISH dots, while larger probes secure high sensitivity of detection. Probe homogeneity is especially relevant for large-size probes, given that an uneven distribution of oligos in the probe might cause the signal to split into two or more signals, which might be undesirable for certain applications. Importantly, our probe design interface runs automatically in the cloud, and can use as input any available oligo database, including all those created with OligoMiner. As such, iFISH4U provides a highly versatile and user-friendly environment for designing new DNA FISH probes against multiple genomes of interest.
4. We have invested resources into generating a new database of 20-mers orthogonal to the human genome from the existing database of 25-mers (Xu et al, *PNAS* 2009; PMID: 19171886), which is a nontrivial task that is computationally demanding. The truncation of orthogonal oligomers is not sufficient to ensure the orthogonality of the new shorter sequences, which need to be mapped and compared to the reference genome again. The reason why we opted for 20-mers, instead of 25-mers, was to reduce the length of the sequences in the oligo pools, given that the error rate increases with the oligo length. Moreover, the subset of 20-mers orthogonal to the genome, which we identified, was designed with the goal of minimizing the risk of cross-hybridization between the 20-mers themselves as well as with their reverse complementary sequences, in order to be able to selectively amplify the oligos corresponding to one probe from one complex oligo-pool, as described in our probe amplification protocol.

The method used to turn these sequences into probes quite closely follows published protocols in general conception and specific detail, and it appears to me the few deviations from the published versions increase the cost and complexity and decrease the efficiency rather than the reverse (see below). No quantitative comparison between approaches is described nor any evidence of further optimization of parameters relative to prior schemes.

Given that, as stated above, the main goal of our work was to create an ever-growing repository of high-quality DNA FISH probes, we needed to make several adaptations to the previously published protocols, in order to optimize the large-scale parallelized production of hundreds of probes in 96- or 384-well plates. We have now also made our protocol compatible with a high-throughput liquid dispensing device (I-DOT, Dispensix) and provide a comprehensive **Step-by-step protocol** in the Supplementary Information. However, in the main text we clearly state that this protocol is entirely based on the published protocols previously applied to synthesize Oligopaints and MERFISH probes.

Following the Reviewer's suggestion, we have made a systematic side-by-side comparison between our and previously published protocols for oligo synthesis using PCR and *in vitro* transcription. We present this comparison in the **revised Supplementary Cost Analysis**, and show that while the cost of producing a ready-to-use ssDNA probe is almost identical in iFISH vs. MERFISH (\$86.21 vs. \$85.40, respectively), the iFISH protocol is much more cost-effective when producing large quantities of probes as PCR products, which can then be distributed to other researchers, like we set out to do with our repository. We did not, however, perform an experimental comparison between our protocol and previously published protocols, since, as we have explained above, our primary goal was to provide the community with long-needed user-friendly probe design tools and a freely accessible repository of high-quality tested probes, targeting hundreds of loci along the human genome.

Diverse and beautiful applications of these probes are shown in cultured human cell populations, though many of these applications have been previously demonstrated with oligo-pool derived probes synthesized with a method extremely similar in many respects. There does not appear to be a particular set of biological conclusions from these applications aside from their quality.

We thank the Referee for appreciating our work. We agree with the Reviewer that similar applications of oligonucleotide-based probes have already been described before. What we tried to achieve in this work was to adopt a research community-centric approach, and focus on the enabling aspect of our work. Therefore, we decided to only showcase some of the potential applications of iFISH probes, without focusing extensively on a particular application. We deliberately chose, however, to put more emphasis on our approach of studying chromosomal intermingling using chromosome-spotting probes, as we think it is innovative and allows for a more quantitative assessment of this interesting aspect of chromosomal organization than so far possible. We also provide evidence for the existence of cells without any clear chromosome territoriality, which, to our knowledge, is the first of its kind. We believe that those cells would go undetected if they were being studied using what has classically been applied to study chromosome territories, *i.e.*, paint probes. Based on our experience with paint probes, given the extended area that chromosomes occupy in those cells, the FISH signals generated by the paint probes would be considered too dim for further analysis (because of the dilution of the probe, in comparison to cells with more condensed chromosomes). We find this observation very intriguing and worth further investigation, and we think that it nicely exemplifies how our novel approach of visualizing chromosomes can reveal previously unappreciated aspects of chromosomal organization.

The authors eliminate the need for computational oligo selection for applications on human cells, by producing a pre-assembled list of probes for the human genome. Previous work (e.g. Chen...Zhuang, *Science* 2015, Boettiger...Zhuang, *Nature* 2105) relied on a rather convoluted probe design and pipeline which included dependency on some difficult to obtain software from built for microarray design. However, more recent work has already addressed this gap. For example, similar pre-built probe lists for the human genome and genomes of other model systems are already freely available online from the Wu lab at HMS (<https://oligopaints.hms.harvard.edu>). Several open source software packages have been released since which remove the dependence on the less user-friendly software from earlier publications and provide relatively easy to use software for the construction of oligo-pools e.g. (<https://github.com/brianbeliveau/OligoMiner>, <http://zhuang.harvard.edu/merfish.html>) see Moffitt...Zhuang, *Methods Enzymol* 2016, PNAS 2016a, PNAS 2106b; Beliveau...Yin PNAS 2018).

We thank the Reviewer for raising this important issue. While we completely agree with the Referee that recently described open-source software packages, such as OligoMiner, have greatly facilitated the design of oligo-pools, we do not believe that these tools completely solve the challenge of designing DNA FISH probes—especially for researchers without bioinformatics expertise—in the same way as, for example, many open-source, web-based tools allow designing PCR primers inside a target of interest. Importantly, the fact that—to the best of our knowledge—OligoMiner (OM) is currently the only set of oligo databases available for download, should not prevent from developing more inclusive tools, especially given the relatively low coverage of the OM hg19 databases. This is clearly demonstrated in the side-by-side comparisons between our iFISH 40-mers database and the OM hg19 databases shown in the **new Figure 1c-d and Supplementary Fig. 1**.

Moreover, although OM databases can be freely downloaded, there is currently no web-based, user-friendly tool that allows sub-selecting oligos from these databases, given a list of genomic coordinates and a set of optimal probe design criteria. As already mentioned above, the iFISH4U probe design interface that we have now created (<http://ifish4u.org/probe-design>) truly eliminates this gap, by allowing not only the selection of oligos falling within one or more specified genomic windows, but also the optimization of the oligo selection process, based on several features that we know are key to make a probe a ‘good probe’ (probe size, centrality, homogeneity). As iFISH4U can work with any database of oligo sequences (including those designed with OligoMiner), and that it designs probes remotely in the cloud, without the need for

a dedicated bioinformatician, we believe that it represents an important step forward towards making DNA FISH truly at reach for everyone.

The details of the probe design describe here are very similar to previously published approaches (e.g. Chen...Zhuang, Science 2015, Boettiger...Zhuang Nature 2016): 40 nt, non-overlapping, limited GC range 35-80%, homopolymer stretch of 5, similarity of 70% to other genomic regions removed, primers/adapters selected from the same previously published 25-mers and truncated to 20mers. I don't think this is a barrier to publication, but it would be nice to see results from a rigorous test of which of these parameters matter and how much they influence quantitative properties of the assay, rather than presenting a close application of previously parameters. Are the minor differences in these designs improving the staining quality (e.g. probe uniqueness being screened on a 70% similarity instead of rejecting off-target based on a Tm differential or a 16 continuous base match described in prior work, 40mers instead of 42mers etc.)? To argue these are distinct a head-to-head comparison should be presented, though it does not appear that the current manuscript is claiming per se to achieve superior labeling or to use a meaningfully distinct set of parameters.

Inspired by the Reviewer's comment, we have now conducted a systematic review of all the published works describing the design of Oligopaints. As it can be seen in the **new Supplementary Table 1**, although different studies have used somewhat similar design parameters and filters, there is surprisingly little homogeneity between different reports. Moreover, for some reason, the FISH experimental conditions used in these works differ quite substantially between each other. In this context, a truly systematic benchmarking of probes designed according to all the different design strategies so far described in the literature would be very challenging, and we believe it is out of the scope of this work, which instead—as stated above—aims primarily at providing useful resources for a broad community of researchers. In the revised manuscript, we now refrain from emphasizing the design strategy for our 40-mers database, by moving this part to the Methods section, and instead highlight our newly developed iFISH4U pipeline, which we believe represents the true novelty and advancement of this work.

In addition to the above, we would like to point out that, despite being very curious about the actual contribution of the different parameters to the efficacy of the oligos/probes, our approach in generating the current database was to be as inclusive as possible. This means that, even if we saw that certain parameters are more desirable than others, in practice we would be driven by the need to identify as many potentially useful oligos as possible, in a given region of interest, even if this would mean to include oligos of relatively poor quality in our probes (this is because it is the mass-effect of having multiple oligos bound to the same region that produces the signal, while a single oligo binding unspecifically to an off-target will not have a major impact on the observable FISH signal). The success of this approach can be seen in the comparison we made between iFISH and OM probes, as shown in the new **Supplementary Fig. 2a-e**. Despite us being more inclusive in designing our 40-mers database (for example, by allowing a broader range of the GC content, as shown in the plot below), our probes yielded brighter signals, even though all the probes contained the same number of oligos.

In line with this, in the future we plan to expand our databases (and provide all of them through our iFISH4U platform), by creating a new set of databases with decreasing stringency, in order to be able to cover more and more regions. We also plan to systematically assess the effect of having a larger or smaller fraction of oligos with suboptimal characteristics on the FISH signal quality.

Finally, motivated by the Reviewer's suggestion, we have examined the possible effect of different features—such as DNA accessibility and gene expression— on the overall quality of FISH signals, for which no available data exists. To our surprise, neither the DNA accessibility nor the expression of the genomic targets of the 153 probes, which we individually tested, seemed to influence the corresponding signal-to-noise ratio (SNR). We have included these new data in the **new Figure 2c and 2d**.

This seems to be a missed opportunity. I suspect multiple labs have collected some data on 30mers vs 40mers, or effect of limited GC-range, the effect of excluding homopolymers, but I am not aware that data is published nor that probe designs optimized for RNA FISH are optimal for DNA FISH and v.v. or whether the probe design and hybridization protocol if co-optimized would come to the same value (aside from the Xu...Elledge microarray binding screen of DNA oligos, which is not in vivo and may differ substantially from binding on cells in traditional FISH buffers).

We have leveraged on the fact that we have now individually tested 153 probes, and used the fact that they differ in their average GC-content in order to study whether the SNR changes depending on the GC-content of a probe. As it can be seen in the plots below, we did observe a trend for probes with an average GC-content above 60% to be associated with a worse SNR. However, since this effect was statistically significant only in the case of probes labeled with the Alexa Fluor 594 dye, we decided to refrain from including these data in the revised manuscript, and instead only present them here.

The probe synthesis protocol does not deviate substantially from published protocols (e.g. Chen...Zhuang, Science 2015), with the notable exception of a few added steps on purification and quantification of RNA that are not explained nor contrasted with prior methods. As these added steps take time, require mildly expensive reagents (magnetic beads, a sorting magnet, QuBit reagents and system) and result in some loss of sample material, it appears to me that the previously published protocol was cheaper, faster, easier and more efficient. Previous reports showed that RNA purification is dispensable in this protocol, and that the RT reaction can be run with saturating amounts of primer to ensure maximum conversion to ssDNA (removing the need for precise quantification prior to the RT step). Excess primer from the saturating reaction is removed in the oligo-cleanup. Given the emphasis on cost, ease of use, and speed, these modifications are confusing. Purification is likely to result in some loss of product. QuBit quantification is also a non-trivial cost in probe synthesis and sacrifices a small amount of product and time. If the authors' analysis has revealed an advantage of this additional step (which adds cost and complexity to the protocol) it would be nice to see that data presented in the manuscript.

The reason why we added a step of RNA purification, after IVT, is that, in this way, we can measure the concentration of RNA, so that we can use the same amount of RNA for each probe. This is important to perform the RT step efficiently and to make the yield reproducible across multiple probes synthesized in parallel, as well across different experiments. Regarding the cost-effectiveness of this approach, as shown in the plot below, if the maximum possible amount of RNA is used in a single RT reaction, the cost per probe by adding the RNA purification step is only moderately higher compared to the cost without RNA purification (~\$1.9 vs. ~\$2.3 per probe). However, as soon as the RNA input in the RT reaction is inferior to the maximum possible, the cost per probe sharply increases (see plot below). For example, since up to 5 μg of RNA can be used in a standard RT reaction, if less than 5 μg is used, the cost of producing ssDNA becomes higher. If more than 5 μg is used, it means that some RNA is wasted because surplus RNA cannot be converted to ssDNA. We now present these considerations in the **revised Supplementary Cost Analysis**.

Another minor difference with potentially negative impact on the protocol is the use of RNase OUT in place of RNasin Plus as the RNase inhibitor in the RT reaction. Is RNase OUT as stable as RNasin Plus at 50C? Many RNase inhibitors lose efficacy at 50C, which results in loss of product.

According to the RNaseOUT user's manual "RNaseOUT™ Ribonuclease Inhibitor is compatible with all enzymes used in RT-PCR. It has been used with the Elongase® Enzyme Mix in long RT-PCR mixture (40 units/20 μL of reaction mixture)". This means that RNaseOUT can be active at 50 C using the Maxima H RT transcriptase, which we use in the iFISH probe production protocol. Further evidence that RNaseOUT is active in RT-PCR at 50 C can be found in Elfering et al., JBC 2002 (PMID: 12154090).

While it is valuable to confirm the efficiency of these methods outside of the lab and collaborator teams which developed them, there is clearly much room to explore which would substantially enhance the impact of the paper. For example: Does non-overlapping probes actually help? Would overlapping probes work better? (It is unlikely all the probes stick to their targets in all the cells, given the spot-detection rates for small batteries of probes).

We very much share with the Reviewer the curiosity about the effect of designing probes by including overlapping oligos, on the efficiency of a probe. However, we are afraid that, while such a modification would not dramatically increase the quality of the probes, it would be incompatible with our ambition of creating a large repository of FISH probes, simply because of much higher costs.

Having said that, we fully agree with the Reviewer that it would be extremely valuable if a well-defined set of probes were tested by multiple groups independently. This would represent a significant step towards establishing standardized DNA FISH probes. We believe that, by establishing our open-source iFISH4U platform and repository of probes, we have precisely moved towards this direction, and we sincerely hope that our efforts will contribute to the standardization of DNA FISH approaches for studying 3D genome architecture.

Do all these 'orthogonal primers' work equally well (quantitatively)? The authors own data as well as comments in the community would suggest not – 15 – 33% of probes did not even amplify sufficient for testing in the first round, let alone a quantitative comparison of how abundance, library complexity, and resulting contrast/brightness in FISH images. How many adapters were tested? Do all work equally well?

This is a very good point. We have now quantitatively assessed the oligo complexity and performance of the probes in our repository, using two approaches:

- 1) In the first approach, we investigated the oligo complexity of four probes, randomly chosen among the 330 probes described in our manuscript. To this end, we tested by real-time PCR how many of the 96 oligo-species corresponding to each probe, which are present in the original oligo-pool, were actually amplified and converted to ssDNA during the PCR, IVT, and RT steps. As shown in the **new Supplementary Fig. 3f**, only one oligo in one probe, and three oligos in a second probe were not amplified in three replicate PCR experiments, which corresponds to a drop-out rate of ~1%. For two of the four probes (2.22 on chr2, and 5.16 on chr5), we used as PCR template a pool of all the probes from the same chromosome, produced through a single IVT reaction. Even in this case, the fraction of oligo species that were present in the final probe was higher than 97%, indicating that our pipeline for parallelized probe oligo amplification from a single oligo-pool is very efficient.
- 2) In the second approach, already mentioned above, we undertook a large-scale validation effort and individually tested 153 out of the 330 probes in our repository. For each probe, we computed the mean number of FISH dots per nucleus per cell in at least 100 cells (range: 209–1,939 cells per probe), as well as the mean signal-to-noise ratio (SNR) per cell. Remarkably, as shown in the **new Figure 2a and 2b**, on average 78% of all the cells had between 1 and 2 FISH signals per probe ($78.72\% \pm 7.41\%$, mean \pm s.d.), as expected in the HAP1 cell line examined, and the SNR was also highly homogeneous across all the probes. Considering that these 153 probes tested were produced using many different F-R adapter pairs (153), and were visualized using four different fluorescent dyes, we conclude that our orthogonal 20-mer adapters and parallelized probe production pipeline are highly efficient and specific, with no detectable cross-talk between probes produced from the same oligo-pool using different F and R adapter pairs.

I had also a few technical concerns:

I would have liked to see controls of samples labeled with Myc-RNA only to help evaluate the efficiency of the RNA-DNA FISH protocol that is described. One might expect that some of the RNA probe is washed out during the addition of the DNA probe and denaturation of the sample, which the authors perform after RNA probe hybridization. Evidently this does not entirely remove the signal from this highly expressed transcript (the authors show beautiful double-stained images), but it might reduce the efficiency and accuracy of the otherwise quantitative smFISH approach (e.g. the number of cytoplasmic spots detected might be reduced).

We have now compared the *MYC* RNA counts obtained by performing smFISH alone vs. smFISH simultaneously with DNA FISH. As shown in the **new Supplementary Fig. 2i**, the distributions of dots per nucleus are virtually undistinguishable, indicating that performing DNA FISH together with smFISH using our protocol does not undermine the efficiency of the latter. We note that we have moved these results, together with all the other results related to *MYC*, to the Supplementary Figures, since we wanted to use the main figures to showcase the 330 probes.

It seems the DNA FISH probes will also label nuclear RNAs, unless the probes are explicitly designed to be sense to each gene or the sample is treated with RNase (which is not feasible in the combined experiment). This cross-talk may not matter in all applications, but should at least be commented on.

We thank the Reviewer for this suggestion. We have now analyzed how many oligos fall inside coding vs. non-coding regions, for each of the 330 probes as well as for all the 40-mers in our database. We present these results in the **new Supplementary Fig. 1b and 3c**. We also

mention in the main text the fact that most of the oligos in our database fall outside of coding regions, which is particularly relevant for simultaneous DNA-RNA FISH applications.

The description of the chromatic correction protocol is incomplete, which is particularly problematic for a methods-focused manuscript. The authors report they imaged tetraspec beads before and after imaging cells, but do not describe how they used this data to perform corrections. Presumably they computed the chromatic displacements between all beads, calculating a field correction by computing a 2D polynomial (3rd order? 4th order?) map which corrects these aberrations, and applied this field correction to the corresponding data channels. A 3D map would be better but it cannot be computed from beads restricted to the 2D plane of the coverglass. For this the authors would have done better to affix the beads to the cells, which would distribute them throughout the 3D volume imaged and allow for a 3D correction map to be computed (as is typical for sub-diffraction limit chromatic corrections, see Sigal...Zhuang 2015 or Boettiger...Zhuang 2016 for recent examples). With a Plan-Apo chromatically correcting objective these aberrations should be small (largely <200 nm, increasing radially from the center of objective) and may not be a major concern for larger scale features described in this work. However, the approaches used should be clearly described.

We thank the Reviewer for commenting on this important aspect. We have now better explained the approach that we routinely use for chromatic aberrations correction, in the **Methods** section of our revised manuscript. As a direct response we would like to mention that we use a 2D polynomial in the lateral plane and a constant shift in the axial direction. In XY we typically have very small errors (measured as the distance between pairs of beads from different channels) with mean distances between 10 and 50 nm, depending on the pair of channels. It might be true that we should consider also using a more sophisticated model for the aberrations in z. However, looking again at bead pairs which we have imaged, taking the Z-direction into account, with our current setup the mean errors range between 20 up to 145 nm.

For cell cycle identification -- the described method based on DAPI intensity sounds very approximate. It would be nice to validate the DAPI-based approach with some cell cycle markers. For example, the FUCCI system, or an antibody against one of the more abruptly-changing cell-cycle associated proteins, e.g. Geminin. As the cell cycle claims are not essential to the major claims of the manuscript and the methodology orthogonal to the IFISH presented improving the cell-cycle discrimination is not essential.

We routinely perform DNA staining-based cell cycle profiling, following a method similar to a previously published and validated one, that combines the use of the nuclear area in the z-projection with the DNA stain intensity integral (Roukos et al, *Nature Protocols* 2015, PMID: 25633629). Following the Reviewer's request, we have further validated our approach, by performing EdU incorporation followed by EdU and Cyclin A detection. Specifically, we quantified the integral of the intensity of EdU and Cyclin A (CyA) stainings, for each segmented nucleus, and then divided the nuclei in four groups: $\text{EdU}^{\text{neg}}/\text{CyA}^{\text{neg}}$, $\text{EdU}^{\text{neg}}/\text{CyA}^{\text{pos}}$, $\text{EdU}^{\text{pos}}/\text{CyA}^{\text{neg}}$, and $\text{EdU}^{\text{pos}}/\text{CyA}^{\text{pos}}$. Identification (thresholding) of the positive (pos) or negative (neg) status for the stainings was obvious upon visualization of staining density curves, as shown in the left figure below. Next, we classified the four groups as different cell cycle phases, based on prior knowledge: $\text{EdU}^{\text{neg}}/\text{CyA}^{\text{neg}}$ as G1 phase, $\text{EdU}^{\text{pos}}/\text{CyA}^{\text{pos}}$ as S/G2, $\text{EdU}^{\text{neg}}/\text{CyA}^{\text{pos}}$ as G2, and $\text{EdU}^{\text{pos}}/\text{CyA}^{\text{neg}}$ as late-G2/early-G1 (see right figure below). Approximately 78% of the cells labeled as G1 by our approach, with a 3σ threshold, are stained neither with EdU nor with Cyclin A ($\text{EdU}^{\text{neg}}/\text{CyA}^{\text{neg}}$). The adoption of an even stricter threshold of 2σ (data not shown) results in an even higher percentage of cells identified as G1 (~82%). We note that, with our approach, any cell mislabeled as G2 will be computationally discarded at a later analysis stage, based on the number of FISH signals, which would then be approximately twice as much as the expected one. In summary, thanks to this two-steps selection procedure, we are confident that cells classified as G1 using our approach represent a fairly "pure" population.

Reviewer #2

iFISH is a free resource for large-scale parallelized design and production of DNA FISH probes, by E. Gelali, et al., submitted to Nature Communications

E. Gelali et al. describe 330 oligo-based probes for use in studies of human genome organization via fluorescent in situ hybridization (FISH). As proof-of-principle, they target MYC (1 Mb as well as ~14 kb), then 6 locations on chromosome 18, and, finally, 330 locations (using any of three dyes) spaced either ~10 or ~5 Mb apart across chromosomes X and 1-22. The authors then demonstrate the usefulness of these probes for defining chromosome territories (including the intersection of overlapping territories) as well as labeling mitotic chromosomes, discovering two intriguing patterns – that the two copies of chromosome 17 cluster more than would be expected, and that chromosome territories are less distinct in hESC cells than in IMR90 cells. The images are compelling, the data believable, and the writing accessible. Importantly, a 330-probe library will be of great help to laboratories that currently do not have the expertise or resources to design and/or produce probes on their own.

We are grateful to the Reviewer for her/his appreciation of our work and effort to establish a resource that will facilitate the use of DNA FISH by the research community. Following the comments of this Reviewer, as well as of Reviewer #1, we have now considerably expanded the scope and applicability of the resource originally described in our first manuscript, by developing a freely accessible web-based platform, iFISH4U (<http://ifish4u.org>), which supports two key functionalities:

- 1) Browsing probes in our repository, and requesting ready-to-use PCR products that can be used to quickly amplify the probes in-house (<http://ifish4u.org/browse>).
- 2) Designing new probes (<http://ifish4u.org/probe-design>), by mining our database of 40-mers as well as any other oligo database, including the databases generated with OligoMiner.

Additionally, we have done a major effort to individually test and validate 153 of the 330 probes described in our original manuscript, by visualizing each probe individually in HAP1 cells and measuring the number of detected signals per cell as well as the signal-to-noise ratio. As shown in the new **Figure 2a and 2b**, there was a remarkable homogeneous high-quality across all the probes tested, establishing our repository as the first thoroughly tested open-source collection of DNA FISH probes currently available. We believe that this represents a major advancement towards enabling more research labs to use DNA FISH, and towards creating standardized tools in order to promote increased inter-laboratory reproducibility. Accordingly, we have substantially rephrased the original manuscript, by putting more emphasis on this truly novel resource, and moving the detailed description of our 40-mers database and of the high-throughput probe production pipeline to the Methods section. We hope that the Reviewer will appreciate our efforts.

What is unclear, however, is the magnitude of advance over published methods based on and/or related to the technology called Oligopaints. Specifically, it seems that probes produced by the described method, iFISH, are the same in structure as those produced by Oligopaints. As with iFISH, Oligopaints can also be multiplexed to produce multiple individual probes, amplified by T7 polymerase, indirectly labeled, and, if desired, labeled with different dyes (e.g., Beliveau et al. 2015 Nat Commun, Chen et al. 2015 Science, Murgha et al. 2015 Biotechniques, Boettiger et al. 2016 Nature, Shah et al. 2016 Neuron, Wang et al. 2016 Science, Cattoni et al. 2017, Eng et al. 2017 Nat Meth, Bintu et al. 2018 Science, Szabo et al. 2018 Sci Adv, etc.). There is also significant redundancy with published protocols (most recently, OligoMiner from Beliveau et al. 2018 Proc Nat Acad Sci USA) and databases (Oligopaints) for genomic targets. Very recently, Oligopaint probes have even been used to examine chromosome territories (Rosin et al. 2018 PLOS Genetics).

We agree with the Reviewer that the probe production pipeline which we describe—although introducing several important modifications which make it more cost-effective compared to the

protocol used for generating MERFISH probes (see amended **Cost Analysis** in Supplementary Information) and are needed to parallelize the production of hundreds of probes in 96- or 384-well plates—does not represent a major advancement over previously published protocols. Accordingly, as stated above, in our revised manuscript, we have removed the emphasis on this part, and instead we focus on describing the new iFISH4U interface for designing probes and the collection of probes which we now make available to the community as non-profit repository.

Concerning the ‘redundancy with published [...] databases (Oligopaints)’ mentioned by the Reviewer, we would like to note that the only oligo databases currently available for download are those recently described in the OligoMiner publication (Beliveau et al, *PNAS* 2018: PMID: 29463736), as summarized in the new **Supplementary Table 1**, in which we report a comprehensive review of all the publications reporting the use of Oligopaints or MERFISH probes. Importantly, in response to the remarks on the same topic by Reviewer #1, we have now conducted a systematic *in silico* comparison between our 40-mers database and all the OligoMiner hg19 databases. As shown in the new **Figure 1c and 1d and Supplementary Fig. 1**, our database is superior to the best OligoMiner hg19 database (‘Balanced’) in terms of overall number of oligos, genome coverage, and, most importantly, number of small genomic regions (10–20 kb) which contain at least 96 oligos. As clearly demonstrated by the results of our large-scale testing of 153 probes (see above), as well as based on our extensive experience with high-resolution DNA FISH, probes consisting of such number of oligo species allow detecting regions of less than 10 kb with high specificity and sensitivity. This means that our 40-mers database is more suitable than OligoMiner for applications where one or more loci of 10–20 kb are to be visualized (for example, in chromosome-spotting experiments, such as those presented in our manuscript). However, we would like to reiterate that the probe design interface which we have developed (<http://ifish4u.org/probe-design>) can accept as input both our as well as OligoMiner databases, thus providing great flexibility to the users.

Additional comments are as follows:

a) Page 2 (Abstract): “...there is little recognition of DNA FISH as a powerful tool that per se allows investigating unique aspects of chromosome organization in single cells that cannot be captured by Hi-C.” This statement seems out of place, as many researchers, including those in the Hi-C world, would argue that DNA FISH has made significant contributions to what we know about genome organization.

We apologize for not having been clearer here. What we meant is that, even though DNA FISH has been instrumental to build our knowledge of nuclear and genome architecture in the pre-Hi-C era (as demonstrated, above all, by the pioneering work of Christof Cremer and colleagues), following the advent of Hi-C in 2009, a considerably higher amount of publications have reported the use of Hi-C, rather than DNA FISH, as a primary means to study genome organization. While this is understandably related to the much higher throughput achievable with Hi-C compared to DNA FISH, we also think that DNA FISH could have been more broadly deployed, if resources such as the one which we have now developed existed before. We have now rephrased the main text accordingly, and hopefully better convey our intended message.

b) Page 2: “...DNA FISH has lagged considerably behind 3C methods with respect to the number of research groups that are using it as a routine technique.” This statement is puzzling, as it greatly understates the widespread recognition and use of DNA FISH.

In line with the previous comment, we have now re-phrased this sentence both in the Introduction and in the Discussion.

c) Page 3: Prominent papers producing oligo-based probes that should be cited in the Introduction along with references 3 and 4 would be Navin et al. 2006 (Bioinformatics), Yamada et al. 2011 (Cytogenet Genome Res), and Boyle et al. 2011 (Chromosome Res).

We thank the Reviewer for her/his suggestion, and have added these references accordingly.

d) Results: As noted in the summary statement, it is unclear how iFISH differs from Oligopaints in terms of structure and application. (See papers cited above.)

As we have extensively commented above, the key novelty over prior works is the fact that we provide a comprehensive and freely accessible resource of user-friendly probe design tools and ready-to-use validated probes, which will enable a broad community of researchers to use DNA FISH as a routine technique.

e) Results: As many of the probes correspond to <100 oligos, do the authors have information regarding the potential for skewed representation of the individual species of oligos after rounds of amplification?

This is a very good question, which was also raised by Reviewer #1. Indeed, we have now quantitatively assessed the complexity and performance of the probes in our repository, using two approaches:

3) In the first approach, we investigated the complexity of four probes, randomly chosen among the 330 probes described in our manuscript, by testing, with real-time PCR, how many of the 96 oligo-species that are present for each probe in the original oligo-pool were actually amplified and converted to ssDNA during the PCR, IVT, and RT steps. As shown in the new **Supplementary Fig. 3f**, only one oligo in one probe, and three in another were not amplified in three replicate PCR experiments, which corresponds to a drop-out rate of less than 3%. Notably, for two of the four probes (2.22 on chr2, and 5.16 on chr5) we used as PCR template a pool of either all or five of the probes from the same chromosome, respectively, produced through a single IVT reaction. Even in this case, the fraction of oligo species that were present in the final probe was higher than 97%, indicating that our pipeline for parallelized probe oligo amplification from a single oligo-pool is very efficient.

4) For each probe, we computed the mean number of FISH dots per nucleus per cell in at least 100 cells (range: 209–1,939 cells per probe), as well as the mean signal-to-noise ratio (SNR) per cell. Remarkably, as shown in the new **Figure 2a and 2b**, on average 78% of all the cells had between 1 and 2 FISH signals per probe ($78.72\% \pm 7.41\%$, mean \pm s.d.), as expected in the HAP1 cell line examined, and the SNR was also highly homogeneous across all the probes. Considering that these 153 probes tested were produced using 153 different F-R adapter pairs and four C adapters, and were visualized using four different fluorescent dyes, we conclude that our orthogonal 20-mer adapters and parallelized probe production pipeline are highly efficient and specific, with no detectable cross-talk between probes produced from the same oligopool using different F and R adapter pairs.

f) Page 7: Please provide number of nuclei imaged (n) for Figure 1k.

The previous Fig. 1k is now **Fig. 2g**, and we have added the number of cells analyzed.

g) Page 8: The authors are encouraged to incorporate mention and discussion of Rosin et al. 2018 (PLOS Genetics) into their Introduction and Discussion, respectively, as it examines the intermixing of chromosome territories in *Drosophila*.

We apologize for having missed this relevant work. We have now added a reference to this paper both in the Introduction and in the Discussion, and we discuss our findings on intermingling also in relationship to the findings reported in this paper.

h) Page 9: Regarding Figure 4c, might the authors provide an explanation in the text for the choice of which pairs of chromosomes were studied? Might the format of a matrix be a better way in which to present the results?

We thank the Reviewer for her/his suggestion. Accordingly, we have now plotted the mean mixing index between different chromosome pairs using the 'contact matrix' representation

commonly used for Hi-C data. Regarding the choice of which chromosome pairs were studied, our goal was to cover as many pairs as possible, involving both longer and shorter chromosomes. As the number of all possible pairs and, consequently, the number of experiments needed to visualize them is considerably high (253 pairs, considering all human autosomes and chrX), we necessarily had to decide on a minimum set of pairs to start with. For IMR90, we have now imaged all the autosomes and chrX, and a total of 31 different pairs, whereas in hESCs we have imaged 15 pairs, of which 6 imaged in both IMR90 and hESCs.

i) Page 10: “For more than two decades, DNA FISH has remained a highly specialized technique rarely used outside of clinical genetics laboratories. With the advent....interest in DNA FISH (page 10, Discussion).” As in comments a and c, above, I do not believe this statement accurately portrays the field.

As stated above, we have now removed this sentence from the revised manuscript, and provide a more neutral statement reflecting our intended message.

j) Page 11: The Discussion contrasts iFISH to MERFISH, pointing out how, in iFISH, a) the library can be used to generate many individual probes, b) the identity of each probe is encoded in the adapter, and c) the dye for iFISH is introduced during amplification. Thus, authors are directed to Beliveau et al. 2012 (Proc Nat Acad Sci USA) and 2015 (Nat Commun) and Boettiger et al. 2016 (Nature) for comparable protocols in terms of the generation of individual probes from multiplexed libraries, use of barcodes (adaptors), and introduction of dyes during amplification.

We have now removed this part from the Discussion and, as stated above, rephrased the main text to emphasize the resource nature of our work, rather than focusing on the pipeline used for producing a large number of probes in parallel, which, as rightly pointed out by both this and Reviewer #1, does not substantially differ from the methods previously described to synthesize Oligopaints and MERFISH probes.

Reviewers' Comments:

Reviewer #1:

Remarks to the Author:

The authors have addressed my primary concerns.

The revised narrative, additional data, and especially the website substantially strengthen this work as a resource for the community.

I have a few final suggestions where some of the data and analyses that were presented in the response letter could be helpful to the general reader as well. In particular:

I found the new experiments using quantitative comparisons with OligoMiner probes to be very instructive, and personally would encourage the authors to add this data to the supplement rather than just the response letter, though this is their choice.

I also found the discussion of the limitations of the low density approach instructive. While some of these limitations are helpfully discussed in the revised introduction, the limitation of the sparse coverage could also be added to the list.

A very minor correction to the claims of the rebuttal "Yet, the list of barcodes that have been reportedly used to create Oligopaints is very short (at most, 17 different 20-mer barcodes listed in Moffitt et al, PNAS 2016; PMID: 27625426)" I believe Boettiger et al Nature 2016 describe 80+ orthogonal primer sets with Oligopaints in *Drosophila*. The authors' point is well taken though, that a larger set of experimentally validated primers would be especially valuable to the community.

Reviewer #2:

None

Point-by-point response to the Reviewers' comments

Reviewer #1

The authors have addressed my primary concerns. The revised narrative, additional data, and especially the website substantially strengthen this work as a resource for the community.

We would like to thank the Referee for taking time to read our revised manuscript and rebuttal letter, and for providing us with many constructive comments and suggestions throughout the review process.

I have a few final suggestions where some of the data and analyses that were presented in the response letter could be helpful to the general reader as well. In particular:

--I found the new experiments using quantitative comparisons with OligoMiner probes to be very instructive, and personally would encourage the authors to add this data to the supplement rather than just the response letter, though this is their choice.

We thank the Reviewer for appreciating these data. Following her/his suggestion, we have now added them to a new Supplementary Figure 2, and increased the number of Supplementary Figures to seven.

--I also found the discussion of the limitations of the low density approach instructive. While some of these limitations are helpfully discussed in the revised introduction, the limitation of the sparse coverage could also be added to the list.

Accordingly, we have added the following sentence in the introduction:

“One immediate improvement would be increasing the percentage of the genome covered by oligos, given that OligoMiner databases cover genomes relatively sparsely.”

--A very minor correction to the claims of the rebuttal “Yet, the list of barcodes that have been reportedly used to create Oligopaints is very short (at most, 17 different 20-mer barcodes listed in Moffitt et al, PNAS 2016; PMID: 27625426)” I believe Boettiger et al Nature 2016 describe 80+ orthogonal primer sets with Oligopaints in *Drosophila*. The authors' point is well taken though, that a larger set of experimentally validated primers would be especially valuable to the community.

The Reviewer is correct in stating that Boettiger et al (Nature 2016; PMID: 26760202) present more than 17 orthogonal barcodes, in fact 65 forward and 2,793 reverse barcodes. However, such barcodes are orthogonal to the *D. melanogaster* genome, with variable length and high sequence homology among reverse primers (some with only a single-base difference). As such, this list of barcodes is not compatible with DNA FISH on human cells. Moreover, given the high homology between specific pairs of reverse barcodes, their applicability is subordinated to the specificity (low homology) among forward barcode sequences.